# The Phytochemical Profile of the Petroleum Ether Extract of *Purslane* Leaves and Its Anticancer Effect on 4-(Methylnitrosamino)-1-(3-pyridyl)-1-buta-4 None (NNK)-Induced Lung Cancer in Rats

**DOI:** 10.3390/ijms252313024

**Published:** 2024-12-04

**Authors:** Asmaa S. Abd Elkarim, Safaa H. Mohamed, Naglaa A. Ali, Ghada H. Elsayed, Mohamed S. Aly, Abdelbaset M. Elgamal, Wael M. Elsayed, Samah A. El-Newary

**Affiliations:** 1Chemistry of Tanning Materials and Leather Technology Department, National Research Centre, 33 El-Bohouth St., Dokki, Giza 12622, Egypt; asmaa_nrc@yahoo.com; 2Hormones Department, National Research Centre, 33 El-Bouhoths St., Dokki, Giza 12622, Egypt; rise_sun_1982@hotmail.com (S.H.M.); almardeyah@gmail.com (N.A.A.); ghadanrc@yahoo.com (G.H.E.); 3Stem Cells Lab, Centre of Excellence for Advanced Sciences, National Research Centre, 33 El-Bohouth St., Dokki, Giza 12622, Egypt; 4Department of Animal Reproduction and Artificial Insemination, National Research Centre, 33 El-Bohouth St., Dokki, Giza 12622, Egypt; mohamedaly_nrc@yahoo.com; 5Department of Chemistry of Microbial and Natural Products, National Research Centre, 33 El-Bohouth St., Dokki, Giza 12622, Egypt; algamalgene@yahoo.com; 6Chemistry of Medicinal Plants Department, National Research Centre, 33 El-Bohouth St., Dokki, Giza 12622, Egypt; waelmatar@hotmail.com; 7Medicinal and Aromatic Plants Research Department, National Research Centre, 33 El-Bohouth St., Dokki, Giza 12622, Egypt

**Keywords:** *Purslane* leaf, phytochemical profile, lung cancer, HPLC-QTOF/HR-MS/MS

## Abstract

Lung cancer is a prevalent and very aggressive sickness that will likely claim 1.8 million lives by 2022, with an estimated 2.2 million additional cases expected worldwide. The goal of the current investigation was to determine whether petroleum ether extract of *purslane* leaf could be used to treat lung cancer induced by 4-(Methylnitrosamino)-1-(3-pyridyl)-1-buta-4 none (NNK) in rats. In the in vitro extract recorded, promising anticancer effects in A540 cell lines with IC_50_ were close to the reference drug, doxorubicin (14.3 and 13.8 μg/mL, respectively). A dose of 500 mg/kg/day orally for 20 weeks exhibited recovery effects on NNK-induced lung cancer with a good safety margin, where Intercellular Adhesion Molecule-1 (ICAM-1), the lung cancer biomarker, was significantly reduced by about 18.75% compared to cancer control. *Purslane* exhibited many anticancer mechanisms, including (i) anti-proliferation as a significant reduction in Ki67 level (20.42%), (ii) anti-angiogenesis as evident by a considerable decrease in Matrix metalloproteinase-9 (MMP-9) expression (79%), (iii) anti-inflammation as a remarked decline in Insulin-like growth factor 1 (IGF-1) expression (62%), (iv) pro-apoptotic effect as a significant activation in Forkhead box protein O1 (FOXO1) expression (262%), and (v) anti-oxidation as remarkable activation on antioxidant biomarkers either non-enzymatic or enzymatic concurrent with considerable depletion on oxidative stress biomarker, in comparison to cancer control. The histopathological examination revealed that *Purslane* extract showed markedly improved tissue structure and reduced pathological changes across all examined organs caused by NNK. The anti-lung cancer effect exhibited by the extract may be linked to the active ingredients of the extract that were characterized by LC–MS, such as α-linolenic acid, linoleic acid, palmitic acid, β-sitosterol, and alkaloids (berberine and magnoflorine).

## 1. Introduction

Cancers that originate in the lungs, typically in the small air sacs or airways, are referred to as lung cancer. Non-small cell lung cancer and small cell lung cancer are the two primary types of lung cancer. The most prevalent kind of lung cancer is non-small cell lung cancer. More than 80% of cases of lung cancer are related to it [1]. Globally, lung cancer is a highly aggressive and common illness, with an estimated 2.2 million new cases and 1.8 million deaths predicted for 2020. Globally, exposure to tobacco is by far the most significant risk factor for lung cancer; however, country-specific variations in environmental exposures—such as air pollution, arsenic, radon, biomass fuels, and industrial carcinogens—also have an impact on lung cancer mortality and incidence trend [2].

More than 20 recognized lung cancer-specific carcinogens have been found in tobacco smoke, including the tobacco-specific nitrosamine 4-(methylnitrosamino)-1-(3-pyridyl)-1-butanone (NNK) and polycyclic aromatic hydrocarbons [3]. Nicotine itself also has potent potential carcinogenic activity by means of its conversion to nitrosamine compounds, especially nitrosamine ketone and nitrosonornicotine. A small percentage (~10%) of inhaled nicotine appears to undergo endogenous conversion to nitrosamine compounds. Nitrosamine carcinogenicity is believed to be due to a function of increased DNA methylation and, potentially, direct agonist activity on the nicotinic acetylcholine receptor acting to enhance tumor growth, survival, and invasion [3].

*Purslane* (S.N.: Portulaca oleracea L. (Figure 1), Family: *Portulaceae*) is known in Egypt as the Regla plant. Europe, Africa, North America, Australia, and Asia are the native regions of *Purslane*. It grows in semi-arid, dry, warm, and humid conditions. This annual plant features simple succulent leaves, reddish-green or purplish-green stems, deep taproots, abundant secondary roots, and yellow blooms that yield tiny, black seeds. It is an edible plant that has been eaten for hundreds of years as a salad and traditional meal in many nations throughout the world [4].

Around the world, *purslane* is used in numerous ethnopharmacological applications to cure various illnesses. Multiple health benefits of *Purslane* are well-known, including its ability to relieve pain, reduce inflammation, diuretics, prevent fever, stimulate healing of wounds, function as an antioxidant, and fight bacteria and ulcers [4]. Its various pharmacological properties are anti-inflammatory [5,6], antioxidative [7], renoprotective [8], neuroprotective [9], hepatoprotective [10], hypolipidemic [11], muscle-relaxing effects [12], anti-diabetes [13], anti-platelet aggregation [14], anti-Schistosomiasis [15], anti-asthmatic [16], an immunomodulatory [17], anti-acetyl cholinesterase and anti-hepatic fibrosis [18]. *Purslane* is known to have anticarcinogenic properties. Numerous studies demonstrated the anticancer role of *Purslane* in many cancers, such as hepatocellular carcinoma [18,19], colon cancer [20], glioblastoma multiforme [21], ovarian cancer [22], sarcoma [23], lung cancer [19], cervical cancer [24], gastric cancer [25], and pancreatic cancer [26]. *Purslane* contains bioactive compounds with antioxidant properties that act on metastasis and invasion, modulate the immune system, and inhibit tumor formation [22,27]. *Purslane* is rich in specialized metabolites such as alkaloids, flavonoids, cardiac glycosides, saponins, tannins, phenolic acids, and organic acids. Omega-3 fatty acids and β-carotene are both highly concentrated in *Purslane.* Also, *Purslane* contains methoxylated flavone glycosides, O, and C-flavonoids. Cyclo-dopa amides and cerebroside alkaloids were isolated from *Purslane*. Flavonoids, such as quercetin, kaempferol, isorhamnetin, and naringenin, were found in *Purslane* [28]. Based on the previous therapeutic benefits and anticancer effect of *Purslane*, in addition to its beneficial bioactive compounds, we included *Purslane* in the preclinical trial for treating lung cancer.

Maintaining our objective of investigating the anticancer potential of the Egyptian plants in connection to their chemical characteristics [29,30], the current work aimed to (i) unveil the promising potential of *Purslane* leaves petroleum ether extract in combating lung cancer induced by NNK in rats; (ii) study the mode of action of the extract to act against lung cancer; and (iii) identify the extract’s chemical profile by HPLC-QTOF/HR-MS/MS.

## 2. Results and Discussion

Three biomarkers enabled us to prove the ability of the *Purslane* extract to treat lung cancer: (i) relative weight of the lung, (ii) Intercellular Adhesion Molecule-1 (ICAM-1) as a lung cancer biomarker, and (iii) lung histology. In addition, we studied the influence of *Purslane* extract on many vital processes, such as angiogenesis, proliferation, apoptosis, inflammation, and oxidative stress, that are involved in the carcinogenesis process to identify the mode of action of the extract. To find out the side effects of using *Purslane* extract, we studied its effect on the safety profile in diseased and healthy animals.LC-QTOF-HR-MS/MS studied the metabolites of the *Purslane* Pet. ether extract.

### 2.1. The Study on the A549 Lung Cancer Cell Line (In Vitro Investigation)

The neutral red uptake test, which relies on the capability of viable cells to bind and integrate the supravital dye neutral red into the lysosomes, was used to assess the in vitro cytotoxicity activity of *Purslane* extract against the cancer cell line A549 at various concentrations from 25 to 200 µg/mL. As a reference drug, doxorubicin (Dox) had a significant impact on the growth of A549 cells after 48 h when compared to control cells. The viability of A549 cells was not significantly affected by the use of DMSO as a solvent after a 48-h treatment. When compared to the control cells, the examined extract had a significant impact on the inhibition of cell proliferation. The IC_50_ values obtained from the neutral red assay are presented in Figure 2. After 48 h of incubation time, *Purslane* extract with an IC_50_ of 14.3 μg/mL showed more cytotoxic impact, approximately similar to the cytotoxic effect of Dox (IC_50_ = 13.8 μg/mL) in A549-treated cells as compared to control cells. According to these results, *Purslane* extract exhibited the most excellent and most promising cytotoxic effect on A549 lung cancer cells.

### 2.2. The Study on the Experimental Animals (In Vivo Investigation)

#### 2.2.1. The Ability of *Purslane* Leaves Pet. Ether Extracts to Treat Lung Cancer

Three biomarkers enabled us to prove the ability of the *Purslane* extract to treat lung cancer: (i) relative weight of the lung, (ii) Intercellular Adhesion Molecule-1 (ICAM-1) as a lung cancer biomarker, and (iii) lung histology.

The lungs of the cancer group rats swelled and gained weight; therefore, the relative weight of their lungs significantly increased by about 21.25% compared to ve− control (Figure 3A) (*p* ≤ 0.05). In the treated group, *Purslane* extract significantly decreased the relative weight of the lungs by about 25.77%, compared to the cancer group (0.72% ± 0.07 and 0.97% ± 0.01, respectively). *Purslane* extract significantly reduced the relative weight of the lungs of the treated group lower than that of the ve− control: 0.72% ± 0.07 and 0.80% ± 0.01, respectively (*p* ≤ 0.05). There is no significant difference recorded between the relative weight of the lungs of ve− control and ve+ control of the extract.

Serum ICAM-1 of cancer group rats significantly maximized to be about 4.20 times of ICAM-1 of ve− control (1.60 ± 0.01 and 0.38 ± 0.01 ng/mL, respectively) (*p* ≤ 0.05) (Figure 3B). On the other hand, ICAM of treated group rats was significantly minimized (1.30 ± 0.03 ng/mL) in comparison to cancer control (1.60 ± 0.01 ng/mL). In ve+ control, *Purslane* insignificantly increased ICAM levels compared to ve− control.

#### 2.2.2. The Mode of Action of *Purslane* Leaf Pet. Ether Extract in the Treatment of Lung Cancer

Proliferation: Serum Ki67, the proliferation biomarker of the cancer group, was significantly higher than that of the ve− control (8.57 ± 0.50 and 4.71 ± 0.37 ng/mL, respectively), *p* ≤ 0.05 (Figure 4A). Treatment by *Purslane* showed a remarked decrease in proliferation as a significant reduction in Ki67 (6.82 ± 0.17 ng/mL) in the treated group compared to cancer control (*p* ≤ 0.05). In addition, the proliferation of ve+ control was decreased as an insignificant decrease in Ki67 level in comparison to ve− control (4.57 ± 0.41 and 4.71 ± 0.37 ng/mL, respectively).

Apoptosis: Lung cancer induction considerably reduced FOXO1 gene expression by about 55% than that of ve− control (*p* ≤ 0.05) (Figure 4A). *Purslane* extract revealed a pro-apoptotic effect via a significant elevation in expression levels of the FOXO1 apoptotic gene by about 262% for the extract compared to cancer groups (*p* ≤ 0.05) (Figure 4A).

Angiogenesis: NNK administration significantly elevated MMP9 gene expression (64%) with respect to ve− control. *Purslane* extract showed an anti-angiogenic effect, as a considered decrease in the angiogenic gene, MMP9 (by about 79%) relative to the cancer control (*p* ≤ 0.05) (Figure 4A). On the contrary, compared with the ve− control group, *Purslane* extract revealed a remarkable decrease in MMP9 production (*p* ≤ 0.05) (Figure 4A).

Inflammation: Lung cancer induction caused inflammation, with a significant increase in the IGF-1 gene (about 73%) in comparison with ve− control. *Purslane* extract caused inflammatory inhibitory action as a remarked decrease in IGF-1 expression levels (by about 62%) in comparison with cancer control (*p* ≤ 0.05) (Figure 4C). *Purslane* extract caused an insignificant reduction in IGF-1 expression in contrast with ve− control group (*p* ≤ 0.05) (Figure 4C).

Oxidative stress: *Purslane* extract exhibited remarked antioxidant characteristics, showing by considered elevation in glutathione L reduced (GSH) level and considered activation on GSH-related enzymes (glutathione reductase, GR; glutathione S-transferase, GST; glutathione peroxidase, GPx), catalase (CAT), and superoxide dismutase (SOD) concurrent with a substantial reduction in lipid peroxidation (MDA) of lung tissue of ve+ control, compared to ve− control (*p* ≤ 0.05), Figure 5.

In the cancer control group, all antioxidant biomarkers were depleted. Compared to ve− control, the lungs of the cancer control group appeared with low levels (3.63 ± 0.13 mmol/g lung tissue) and inactivated antioxidant enzymes: GR (3.01 ± 0.1 U/g lung tissue), GST (2.38 ± 0.08 U/g lung tissue), GPx (1.80 ± 0.06 U/g lung tissue), CAT (1.60 ± 0.24 U/g lung tissue), and SOD (4.81 ± 0.42 U/g lung tissue). Also, the lungs of the cancer control rats suffered from lipid peroxidation as a high amount of MDA (11.23 ± 0.37 nmol/g lung tissue), compared to ve− control (*p* ≤ 0.05).

In comparison to the cancer control, force-feeding *purslane* extract for 20 weeks significantly elevated GSH concentration (6.68 ± 0.12 mmol/g lung tissue) and enzyme activities: GR (4.90 ± 0.35 U/g lung tissue), GST (4.04 ± 0.2 U/g lung tissue), GPx (2.98 ± 0.17 U/g lung tissue), CAT (6.02 ± 0.05 U/g lung tissue), and SOD (5.78 ± 0.18 U/g lung tissue). Also, the MDA concentration of the treated group (8.44 ± 0.73 nmol/g lung tissue) was dramatically reduced compared to cancer control.

The meaning of each value is the mean of six repeats. The one-way ANOVA test was used to examine the data in order to compare means at *p* < 0.05. Values that have the same superscript letter do not differ substantially from values that have different letters at *p* ≤ 0.05.

#### 2.2.3. The Safety Profile Study of *Purslane* Leaf Pet. Ether Extract on the Experimental Animals for 20 Weeks

*Liver functions*: Administration of *Purslane* extract for 20 weeks improved the liver performance of the ve+ control to be healthier than the ve− control. In ve+ control, total protein, albumin, and globulin were close to those of ve− control; meanwhile, the AST and ALT activities were lower than ve− control (Figure 6A,B).

On the other hand, cancer group rats appeared to have disrupted liver performance. In the cancer group, total protein production and its fractions, albumin, and globulin, were depleted significantly by about 27.26, 26.48, and 27.86%, respectively, in comparison to ve− control (*p* ≤ 0.05). Additionally, ALT and AST were significantly more activated in the cancer group than ve− control (31.70 and 21.54%, respectively). On the contrary, the liver performance of the treated group rats was healthier than that of the cancer group. *Purslane* extract significantly improved total protein and its fraction; albumin and globulin in the treated group were more significant than those of the cancer group by about 22.98, 39.64, and 9.85%, respectively. Also, it significantly decreased AST and ALT release into the bloodstream by about 42.57 and 30.08%, respectively, relative to the cancer group. Interestingly, *Purslane* treatment returned the liver functions of the treated group to normalization, which was close to ve− control.

The mean of six replicates is included in each value. The one-way ANOVA test was used to examine the data in order to compare means at *p* < 0.05. Values that have the same superscript letter do not differ substantially from values that have different letters at *p* ≤ 0.05.

*Renal functions*: In ve+ control, administration of *Purslane* extract for 20 weeks ameliorated renal performance to be better than that of ve− control. *Purslane* significantly reduced serum uric acid (16.71%) and urea (4.65%) concentrations compared to ve− control (*p* ≤ 0.05) (Figure 6C).

Lung cancer group rats are characterized by remarked confusion in renal performance. All renal function biomarkers significantly rose as uric acid (26.44%) and urea (14.80) compared to ve− control (*p* ≤ 0.05). *Purslane* stopped this confusion and significantly reduced all elevated biomarkers, serum uric acid (30.60%), and urea (7.36%) concentrations in the treated group compared to cancer control (*p* ≤ 0.05). Interestingly, the uric acid concentrations of the treated group are remarkably depleted, close to those of ve− control group.

#### 2.2.4. Histopathological Study

##### Gross Pathological Observations

The examination revealed the presence of numerous small and large nodules in the lung tissue.

##### Histopathological Examination

The data presented suggest that 20 weeks of NNK exposure in rats can induce various lung tumors and various forms of lung cancer, including carcinoma, in a time-dependent manner. In both negative control and *Purslane*-treated groups, histopathological examination of lung tissue sections showed no significant alterations, with typical alveolar architectures and standard interalveolar septa (Figure 7A–D), respectively. Nonetheless, the lung cancer control group that got repeated doses of NNK showed mild to severe hyperplasia, adenoma, and adenocarcinoma. Most bronchioles were necrotic and deteriorated, with significant interstitial macrophage infiltration in the alveolar spaces. Some sections under examination showed alveolar macrophages, inflammatory cell infiltrations, alveolar hemorrhages, emphysema, squamous cell carcinoma, and adenocarcinoma, in addition to the proliferating cells exhibiting cellular atypia, squamous metaplasia, and squamous dysplasia (Figure 7E,F). Alveolar structures were greatly absent; on the other hand, lung tissue sections from the group treated with *Purslane* ether extract demonstrated improved lung structures, lessened alveolar dysplasia, hyperplasia of the lining epithelial lining, with slightly congested blood vessels, and mild macrophage infiltration **(**Figure 7G,H).

In the histopathological examination, the kidneys of the control negative group, as well as those treated with *Purslane* ether extract, displayed a normal renal architecture with intact and well-preserved renal structures (Figure 8A–D), respectively. The renal cortex exhibited healthy glomeruli with normal cellularity, and the renal tubules appeared unremarkable, with clear lumen. In stark contrast, the kidneys of the cancer control group showed significant pathological alterations. Moderate to severe vacuolar degeneration was observed in the epithelial lining of renal tubules, indicating severe cellular injury. Additionally, there was severe congestion of the intertubular blood vessels and capillaries. Moreover, the renal glomeruli in the NNK-treated group showed moderate congestion along with hypercellularity of the glomerular tuft. The presence of moderate focal renal hemorrhage was also noted, suggesting that the structural integrity of renal vasculature was compromised (Figure 8E,F). In contrast to the severe pathological changes observed in the NNK-treated group, kidney tissue sections from the group that received *Purslane* ether extract after being subjected to successive toxic doses of NNK showed a substantial reduction in the severity of these alterations. The *Purslane* ether extract appeared to exert a protective effect, leading to marked amelioration of the kidney’s pathological condition (Figure 8G,H).

In the liver histopathological examination, the control negative group, as well as those treated with *Purslane* ether extract, displayed a normal uniform chord-like architecture (Figure 9A–D), respectively. In contrast, after 20 weeks of NNK exposure, the treated liver sections exhibited remarkable pathological changes. Figure 9E,F reveal a disorganized architectural pattern with focal areas of necrosis and multifocal apoptosis. Massive steatohepatitis was also noticed, characterized by a widespread number of microvesicular lipid droplets that appeared as small cytoplasmic vacuoles. These pathological findings collectively underscore the severe impact of NNK exposure on liver morphology and cellular integrity. The *Purslane* ether extract seemed to exert a protective effect, leading to marked amelioration of the liver pathological condition, where the hepatic chord architecture showed marked restoration, indicating tissue recovery, and also reduced steatosis and inflammation observed as compared to the NNK-treated group, suggesting a potential shift toward healing or adaptation in the hepatic tissue (Figure 9G,H).

### 2.3. The First Chemical Profile of a Pet. Ether Extract from Purslane Leaves by HPLC QTOF/HR-MS/MS

#### 2.3.1. HPLC-QTOF/HR-MS/MS

The primary purpose of this study was to investigate *Purslane*’s secondary metabolome in order to establish a framework for determining its possible uses. In order to achieve this goal, the secondary metabolites of *Purslane* were profiled using a combination of Pet. ether extraction and HPLC-QTOF/HR-MS/MS. The chemical components of *Purslane* were investigated in both positive (+ve) and negative (−ve) ionization modes (Appendix A). We showed that this plant included a large number of alkaloids, fatty acids, and steroid components. We first detected them in the Pet. ether extract of *Purslane*.

The total ion current (TIC) and base peak MS-chromatograms (BPCs) in both −ve and +ve ionization modes (Figure 10 and Figure 11) showed that the alkaloid and fatty acid constituents were abundant in *Purslane* and known to ionize more readily in (+ve) mode than fatty acids, which ionize better in −ve mode [31]. The MS/MS spectra of both −ve and +ve ions originated from anions or cations, specifically [M−H]^−^ and [M+H]^+^, respectively. Positive (+ve) ionization generally revealed more noticeable peaks in *Purslane* and had superior sensitivity as compared to negative MS/MS.

Using HPLC-QTOF/HR-MS/MS with (+ve) and negative (−ve) dual ionization modes, compounds in *Purslane* were tentatively identified based on ionization modes (−ve and +ve), retention times (R_t_), exact mass detected, chemical formula, and an error ppm (between actual mass and observed mass) for all phytoconstituents. By comparing the published data with the spectra of reference compounds, the limit of detection for each peak of the compounds was determined using the MS/MS fragment ions.

A total of 53 chromatographic peaks were identified, including 26 alkaloids (e.g., jatrorrihizin), 2 terpenes (e.g., Nigellic acid), 18 fatty acids (e.g., linolenic acid), and 7 sterols (e.g., stigmasterols) listed in Table 1.

#### 2.3.2. Characterization of Highly Abundant Alkaloids

##### Isoquinoline Alkaloids

Isoquinoline alkaloids are among the most abundant and structurally various natural components. They are known to have a wide range of biological properties and have gained interest in medication development. These substances are biogenetically produced from phenylalanine and tyrosine, with an isoquinoline or tetrahydroisoquinoline ring serving as a basic structural feature in their skeleton (Figure 12) [32].

The positive ion spectrum (MS/MS) of alkaloid peaks showed greater sensitivity and more discernible peaks than the negative-ion mode. All of the isoquinoline alkaloids produced strong M^+^ or [M+H]^+^ ions [33]. According to the data in Table 1, a variety of quasi-molecular ions were identified for various alkaloids in (+ve) ion mode. Consequently, jatrorrhizine (**4**), stepharanine (**6**), dehydrocorydalmine (**9**), berberine (**12**), columbamine (**14**), and palmatine (**21**) were recognized as protoberberine-type alkaloids. It was determined that magnoflorine (**5**) and menisperine (**17**) were aporphine-type alkaloids. Tembetarine (**3**) and colletine (**8**) were identified as alkaloids of the benzylisoquinoline class. The following were identified as amino alkaloids: caffeoyltyramine (**11**), coumroyltyramine (**13**), feruloyltyramine (**15**), feruloylglycine (**18**), feruloyloctopamine (**20**), and N-[(4-hydroxy-3-methoxyphenyl)-methyl]-dodecylamide (**25**) [34,35].

##### Aporphine-Type Alkaloids

In magnoflorin (**5**), a molecule of dimethylamine (CH_3_)_2_NH is the initial loss observed, followed by CH_3_OH, CO, C_2_H_4_, and H_2_O as the most common pathways. In the most common pathway, there are two possible paths for the loss of CH_3_OH, one (I) between C-1 and C-2 and the other (II) between C-10 and C-11. Pathway II appears to be more desirable because the loss of H_2_O in subsequent fragmentations is easily described when OH is at C-1 (Figure 1).

The presence of the two pairs of vicinal OH and OCH_3_ groups in magnoflorin increases the possibility of losing two molecules of CH_3_OH in sequence as the primary mechanism. The steric interactions between the epoxy group produced after the neutral loss of CH_3_OH (at C-10 and C-11) and the OH at C-1 may be the reason for the unexpected results (red color) shown in Figure 1. Magnoflorine shows a possible mechanism of these stearic interactions. When compared to the main pathway, the losses of CH_3_ and the second molecules of CH_3_OH were observed at *m*/*z* 282.2715 and 265.2271 as minor fragments in MS/MS and 307.2593 as a major ion produced from the loss of H_2_O and OH groups easily (Appendix A) [36].

Menisprine fragmented similarly to magnoflorine in MS/MS, providing *m*/*z* 310.9110[M^+^-(CH_3_)_2_NH]^+^ and [M^+^-(CH_3_)_2_NH-CH_3_OH]^+^ at *m*/*z* 279.1961, respectively (Figure 2). The ion spectra of [M^+^-(CH3)_2_NH-CH_3_OH]^+^ at *m*/*z* 279 exhibited CH_3_ radical loss rather than CO losses when compared to magnoflorine. This can be explained by the creation of a furan-like intermediate [37]. Due to the oxygen atom’s involvement in an aromatic ring, CO cannot be removed. However, in successive fragmentations, CO losses were noticeable, and unexpected CH_2_O losses were also detected. The appearance of the major ion at *m*/*z* 293.2103 results from the loss of (63 amu); CH_3_OH (32 amu) and OCH_3_ (31 amu) suggested peak (**17**) as menisprine (Appendix A). Magnoflorine and menisprine were tentatively detected and recognized for the first time in *Purslane* based on their MS/MS and compared to their reported data.

##### Protoberberine-Type Alkaloids

Two protoberberine analog alkaloids (peaks **6** and **10**) with the same precursor at *m*/*z* (324.2862/324.2878), the same molecular formula (C_19_H_18_NO_4_**^+^**), and different time elution at (20.418/21.364 min). They generated characteristic fragments of protoberberin-type alkaloids at *m*/*z* 309.2202[M^+^-CH_3_]^+^, 294.2925[M^+^-2CH_3_]^+^ and 292.2032[M^+^-CH_3_OH]^+^, respectively. Furthermore, peaks **6** and **10** gave identical [M^+^-H_2_O]^+^ ions at *m*/*z* 306.2829/306.2587, indicating the existence of a hydroxyl group at C-3. The suggested fragmentation mechanism for compound (**6**) is illustrated in Figure 3. As a result, the tentative identities of compounds **6** and **10** were determined to be stepharanine and dehydrodiscretamine [38].

Three isomers with identical [M]^+^ ions were produced at *m*/*z* 338.3384/338.3382/338.3391 (mass error = 1.8/−1.9/−3.7 ppm), and the same molecular formulae C_20_H_20_NO_4_**^+^** were tentatively assigned based on their fragmentation pattern. All of them generated daughter ions at *m*/*z* 307.2313/307.2104/307.2104[M^+^-OCH_3_]^+^,292.2130/292.2049[M^+^-OCH_3_-CH_3_]^+^, 275.2001/275.2061/275.2061[M^+^-OCH_3_-CH_3_OH]^+^, 306.19999, and 320.2450/320.2627 [M^+^-H_2_O]^+^, respectively. The presence of an OH group at C-3 was confirmed by the neutral loss of H_2_O molecules and resulted in a product ion at *m*/*z* 320.2450/320.2627 [M^+^-H_2_O]^+^. The observed product ion at *m*/*z* 322.1979 suggested the presence of two (OCH_3_) groups at C-9 and C-10 (Figure 3). Therefore, these compounds were tentatively detected as jatrorrhizine (**4**), columbamine (**14**), and dehydrocorydalmine (**9**), respectively [39,40,41]. The presence of a high-intensity base peak (*m*/*z* = 338.2137) of anticancer jatrorrhizine at the time of elution 15.075 min (C_20_H_20_NO_4_^+^; error = 0.8) in +ve mode indicated that it was abundant in *Purslane* (Appendix A).

Compound (**12**) generated [M+H] ^+^ ion at *m*/*z* 336.2517 (C_20_H_18_NO_4_^+^, an error ppm = 0.12). A series of product ions was obtained at *m*/*z* 321.1630[M^+^-CH_3_]^+^, 306.1610[M^+^-CH_3_-CH_3_]^+^, and 320.2473[M^+^-CH_3_-H]^+^, suggesting the possible existence of two adjacent (OCH_3_) groups. Additionally, the major product ion at *m*/*z* 292.0728 [M^+^-CH_3_-H-CO]^+^ was created by the product ion at *m*/*z* 320.2473. Furthermore, the ion at *m*/*z* 334[M^+^-2H]^+^ indicated the presence of C-C single bonds at C5 and C6, possibly due to a more stable π-conjugated system created by the loss of [M^+^-2H]^+^, so compound (**12**) was identified as berberine based on a comparison to the reference standard [33].

The metabolite (**21**) exhibited the [M]^+^ ion at *m*/*z* 352.2412 and (C_21_H_22_NO_4_^+^, an error ppm = −0.6 ppm). The MS/MS spectra identified *m*/*z* 335.2989 [M^+^-CH_3_-2H]^+^ as the base ion peak. Other daughter ions at *m*/*z* 337.2721 [M^+^-CH_3_]^+^, 323.9123 [M-CO]^+^, and *m*/*z* 307.9925 [M^+^-CH_3_-H-CO]^+^, were also detected. By comparison with the literature data, compound (**21**) was tentatively assigned as palmatine [42].

##### Alkaloids of the Benzylisoquinoline Class

The two highly abundant isomers that had not been previously identified in the Pet. ether extract of *Purslane* produced their [M]^+^ ions at *m*/*z* 328.1046 (C_20_H_26_NO_3_^+^, an error ppm = 0.7) and 344.3013 (C_20_H_26_NO_4_**^+^**, an error = 0.6 ppm), respectively. In MS/MS analysis, one generated product ions [M^+^-OH]^+^, [M^+^-CH_3_OH]^+^, [M^+^-OCH_3_-H_2_O]^+^, [M^+^-OH-2CH_3_]^+^ [M^+^-(CH_3_)_2_NH]^+^, [M^+^-(CH_3_)_2_NH-H_2_O]^+^, [M^+^-CH_3_-H_2_O-2H]^+^ and [M^+^-(CH_3_)_2_NH-CH_3_OH]^+^ at *m*/*z* 311.2313, 296.3214, 279.2029, 283.1830, 281.1769, 265.1799, 293.2477 (the most abundant ion), and 251.2050 (Figure 4) proposed as colletine (**8**). Furthermore, an (OCH_3_) group and a (CH_3_OH) molecule were lost by the colletine product ion at *m*/*z* 175 to produce the main ions at *m*/*z* 144 and *m*/*z* 143, respectively.

Similarly, metabolite 3 produced characteristic daughter ions at *m*/*z* 327.0811[M^+^-OH]^+^, 299.1520[M^+^-(CH_3_)_2_NH]^+^, 267.2640[M^+^-(CH_3_)_2_NH-CH_3_OH]^+^, and 235.2196[M^+^-(CH_3_)_2_NH-CH_3_OH-CH_3_OH]^+^, suggested for tembetarine (Figure 5). The pathway of a very useful daughter ion of tembetarine appeared at *m*/*z* 299, they are in competition with one another because of skeleton fragmentation peripheral substituent losses. Using an a-cleavage fragmentation at the 8, 9-position of the [M]+ ion, tembetarine produced the ions at *m*/*z* 207 and 137.

##### Amino Alkaloids

Five amino-alkaloid constituents tentatively identified for the first time in Pet. ether extract have identical pathways for fragmentation and identical product ions (Figure 6). Additionally, hydroxycinnamic acid amide fragmented similarly, expelling a tyramine molecule and giving the base ions at *m*/*z* 147.0994[M+H-C_8_H_10_NO^−2^]^+^ (coumaroyl), *m*/*z* 163.10785 [M+H-C_8_H_10_NO^−2^]^+^ (caffeoyl), *m*/*z* 179.1451 [M+H-C_8_H_11_NO-]^+^, and 177.1267 [M-H-C_2_H_4_NO_2_]^−^ (feruloyl). By removing the tyramine molecule (137 amu), successive fragmentation was generated by the loss of the CO molecule from the base peak. Furthermore, the tyramine moiety lost NH_3_, resulting in an ion at *m*/*z* 121. Based on fragmentation behavior, compounds (**13**, **11**, **15**, and **20** & **18**) were identified as coumaroyltyramine, caffeoyltyramine, feruloyltyramine, feruloyloctopamine, and feruloylglycine, respectively [43].

##### Cyclodopa Amide Alkaloids

*Purslane* Pet. ether extract contains five cyclodopa amides that have been tentatively identified for the first time and attributed to peaks **2**, **7**, **22**, **24**, and **26**; their structures are drawn in (Figure 13).

#### 2.3.3. Tentative Identification of Highly Abundant Fatty Acids in Pet. Ether Extract from *Purslane*

During the second part of the chromatographic run, negative ionization MS/MS identified several fatty acids (17 FAs) as abundant peaks. The negative ion MS/MS spectra revealed 12 unsaturated FAs and 5 saturated FAs eluting with a higher proportion of organic solvent. Most of them were tentatively identified from extra losses of H_2_O molecules (−18 amu) and CO_2_ moiety (−44 amu) [44,45].

Linolenic (**35**), linoleic (**40**), eicosapentaenoic acid (**44**), arachidonic (**38**), trihydroxy-octadecadienoic acid (**34**), hydroxy octadecadienoic acid (**29**), hydroxy octadecenoic acid (**30**), dihydroxy methyl octadeca-9,12-dienoate (**45**), trans-11-methyl-12-octadecenoic acid (**46**), dihydroxy octadecatrienoic acid (**37**), ricinoleic (**31**), and oleic acid (**43**) were easily interpreted in −ve mode, yielding exact masses at *m*/*z* 277.2161, 279.2303, 301.2165, 303.2290, 295.2259, 297.2425, 297.2434, 293.2467, 295.2641, 309.2042, 327.2889, and 281.2461, respectively. Notice the presence of a difference in mass between the peaks shown above by 2 units, indicating the presence of an additional double bond [30].

Also, MS/MS signals suggested for saturated fatty acids with the lowest polarity [46] and different elution times were detected in peaks **33**, **36**, **41**, and **42** exhibiting an accurate masses at *m*/*z* 326.9309, (an error ppm = 0.82, C_20_H_39_O_3_**^−^**) for 2-hydroxy eicosanoid acid, 283.2650 (an error ppm = 2.3, C_16_H_35_O_2_**^−^)** for stearic acid, 255.2297 (an error ppm = 2.1, C_16_H_31_O_2_**^−^**) for palmitic acid, and 271.2281 (an error ppm = 1.3, C_16_H_31_O_3_**^−^**) for 2-Hydroxypalmitic acid.

MS/MS spectral data of most hydroxylated FAs revealed the loss of 1 or 2 molecules of H_2_O (2 × 18 amu), indicating the existence of extra OH groups as in hydroxy octadecadienoic acid, ricinoleic acid, 2-hydroxy eicosanoid acid, hydroxy octadecatrienoic acid, and 2-hydroxy palmitic acid. This is evidenced by the appearance of their characteristic product ions at *m*/*z* 277.2164[M-H-H_2_O]^−^, 259.2061[M-H-2H_2_O]^−^; 279.2327[M-H-H_2_O]^−^, 261.2215[M-H-2H_2_O]^−^; 291.2029[M-H-2H_2_O]^−^; 275.2055[M-H-H_2_O]^−^, 257.1020[M-H-2H_2_O]^−^; 253.1769[M-H-H_2_O]^−^, and 235.1970[M-H-2H_2_O]^−^, respectively.

Because of their many biological benefits, hydroxylated fatty acids are gaining more attention. Fatty acids that are important for human health: Linolenic acid (omega-3), linoleic acid (omega-6), and hydroxy octadecatrienoic acid were the most abundant unsaturated fatty acids identified for the first time in petroleum ether extract [25,28].

Only one highly concentrated unsaturated fatty acid assigned tentatively only in positive ion mode (+ve) at *m*/*z* 393.2857; molecular formulae (C_20_H_41_O_7_^+^); and Rt = 15.040 min are being recorded for the first time in *Purslane* confirmed as D-glucitol monomyristate (**32**).

#### 2.3.4. Tentative Identification of Sterols in Pet. Ether Extract from *Purslane*

Seven known sterols (Figure 14) were tentatively characterized in (+ve) mode and are being documented for the first time in Pet. ether extract from *Purslane*. Two analogs of cholesterol and cholestanol generated their [M^+^] ions at *m*/*z* 387.2398/389.2447 (an error ppm = 0.7/2.1), (C_27_H_49_O^+^/C_27_H_47_O^+^), respectively. Both of them produced characteristic daughter ions corresponding to the loss of a water molecule (H_2_O) accompanied by successive loss of the CH_3_ group [47]. The MS/MS spectrum exhibited their product ions at *m*/*z* 369.2651[M+H-H_2_O]^+^, 354.4301[M+H-H_2_O-CH_3_]^+^, and 309.2600[M+H-H_2_O-60(4CH_3_)]^+^ confirmed compound (**49**) as cholesterol; 371.2918[M+H-H_2_O]^+^, 356.0695[M+H-H_2_O-CH_3_]^+^, and 309.2895[M+H-H_2_O-60(4CH_3_)-2H]^+^ suggested compound (**48**) as cholestanol. The difference between their base peaks is equivalent to 2 units, proving the existence of a double bond in cholesterol.

Similarly, two analogs with precursor ions at *m*/*z* 401.2657 and 397.3432 gave ions at 383.2570[M+H-H_2_O]^+^, 369.2416[M+H-2H-2CH_3_]^+^, and 355.1109[M+H-H-3CH_3_]^+^ prove compound (**47**) as campesterol; the ions appear at *m*/*z* 379.2682[M+H-H_2_O]^+^, 363.2307[M+H-2CH_3_-4H]^+^, and 349.5022[M+H-3CH_3_-3H]^+^, suggesting compound (**53**) as ergosterol. The difference between their base peaks is equivalent to 4 units, proving the existence of an extra double bond in ergosterol compared to campesterol. Also, *β*-sitosterols (**50**) and stigmasterols (**51**) are other analogs with a difference of 2 units between their base peaks [48]. They generated product ions characteristic of loss of H_2_O. α-Amyrin yielded an [M^+^] at *m*/*z* 427.3549 (C_30_H_51_O^+^, an error ppm = 2.19) was reported in Pet. ether extract from *Purslane* for the first time.

This study demonstrated the anti-lung cancer effect of *Purslane* Pet. ether extract on NNK-induced lung carcinomas in rats. Lung cancer induction caused remarked disruption on several vital processes, including (i) increase in lung cancer biomarker (ICAM) level, (ii) promotion of inflammation (IGF-1 gene), (iii) encouragement of angiogenesis (MMP9 gene), (iv) activation of proliferation (Ki67 biomarker), (v) increase in oxidative stress biomarkers, (vi) decrease in antioxidant biomarkers, and (vii) inhibition of apoptosis in the cancer group compared to ve− control. Pathologically, NNK treatment resulted in severe histopathological alterations in the lung, renal, and liver.

On the other hand, force-feeding *Purslane* Pet. ether for 20 weeks showed considerable amelioration in all these biomarkers towards normalization compared to the cancer group, revealing reduced alveolar dysplasia and improved lung structures, as well as a substantial reduction in the severity of renal and hepatic tissue alterations.

Many studies have investigated the anticancer effect of *Purslane* leaf aqueous and alcoholic extracts in vitro and in vivo models. Most of these studies were performed on in vitro models. Just a few studies have investigated the anticancer effect of alcoholic extract in vivo models. However, the anticancer effect of Pet. ether extract of *Purslane* leaves has not been studied previously: in vitro or in vivo. Our study is the first to examine the anti-lung cancer impact of *Purslane* leaves Pet. ether extract on an in vitro and in vivo scale. The current study was the first advanced preclinical study to investigate the effect of Pet. ether extract on NNK-mediated lung cancer**.**

Section 2 was planned to answer two questions: why and how the *Purslane* extract treated lung cancer.

First, why did *Purslane* extract treat lung cancer? The *Purslane* leaf ether extract demonstrated anti-lung cancer properties because of its chemical makeup, which includes significant amounts of secondary metabolites detected by LC-Mass. The ether extract of *Purslane* leaves contained several anticancer ingredients. The compounds identified in the etheric extract vary between fatty acids, sterols, and alkaloids. The major fatty acids are α-linolenic acid, α-linoleic acid, hydroxy-octadecatrienoic acid, palmitic acid, and 2-hydroxy palmitic acid. Many sterols are identified as stigmasterol, beta-sterol, alpha, and beta-marine. Also, many alkaloids occur in the extract, such as jatrorrhizine, berberine, stepharanine, and tetrahydropamatine, which belong to the isoquinoline alkaloid class, as well as magnoflorine, which belongs to the aporphine alkaloids.

α-linolenic acid, the principal component abundant in our extract, has been demonstrated as an anticancer compound in multiple cancers. Numerous anticancer benefits of α-linolenic acid include antioxidant actions, suppression of tumor metastasis and angiogenesis, apoptosis induction, and inhibition of proliferation [49]. Yang et al. [50] demonstrated that α-linolenic acid suppressed the growth of renal cell cancer cells via significantly elevating proliferator-activated receptor-γ (PPARγ) activity and gene expression and considerably suppressing cyclooxygenase-2 (COX-2). By decreasing the level of phosphorylated ERK1/2 and p38 and increasing the expression of the tumor suppressor proteins Rb and p53, α-linolenic acid prevented the transformation of cervical cancer cells. Inducible nitric oxide synthase (iNOS) mRNA expression was decreased by α-linolenic acid to lower intracellular NO levels. This can limit lipid peroxidation and further activate caspase 3 to produce death. By upregulating the proapoptotic gene Bax, downregulating the antiapoptotic gene Bcl-2, stabilizing hypoxia-inducible factor-1α (HIF-1α), and downregulating fatty acid synthase to increase mitochondrial apoptosis, α-linolenic acid can induce apoptosis [49,51]. By lowering the expression of proteins involved in tumor angiogenesis, such as VEGF, MMP-2, and MMP-9, α-linolenic acid prevents metastasis. High levels of α-linolenic acid have anti-inflammatory properties [52] via the reduction in inflammatory factors. Sala-Vila et al. [53] revealed that by inducing apoptosis through the formation of ROS, α-linolenic acid has the potential to prevent cancer.

Linoleic acid has diverse beneficial effects against cancer. In the mammary fat pads of naked mice, linoleic acid was found to have a detrimental effect on the proliferation of MDA-MB-435 human breast cancer cells and their ability to spread to the lungs [54]. In several human cancer cell lines, linoleic acid was able to activate the PPARγ protein, increase the expression of the PPARG gene, and cause apoptosis. Linoleic acid was able to enhance the expression of PPAR*γ* in non-small cell lung cancer (NSCLC) cell lines (A549 and Calu-1) and alter the mRNA and protein levels of genes involved in apoptosis [55].

Also, hydroxy-octadecadienoic acids (13(S)-HODE) showed an apoptotic effect that was reduced in the presence of a PPAR-γ antagonist [56]. 13(S)-HODE inhibited cell growth in MCF-7 and MDA-MB-231 cell lines and nondifferentiated Caco-2 cells. Moreover, PPAR-γ was down-regulated in response to the 13(S)-HODE administration [56,57]. Yuan et al. [58] reported that the 13(S)-HODE level was considerably decreased in human lung cancer tissue induced by the tobacco smoke carcinogen NNK compared with non-tumor lung tissue. In mouse experiments, 13(S)-HODE started to reduce at 30 weeks after NNK treatment. 9S-HOD exerted cytotoxicity efficacy and induced apoptosis in acute leukemia HL-60 cells [59].

With proven effectiveness against a range of cancers, palmitic acid has become a promising antitumor drug in recent years. Palmitic acid’s main anticancer action is through the mitochondrial route, which triggers cell death. In the G1 phase, palmitic acid also results in cycle stoppage. Additionally, palmitic acid inhibits cell migration, invasion, and angiogenesis, causes programmed cell autophagy death, and works in concert with chemotherapy medications to increase their effectiveness while lowering side effects [60]. Palmitic acid (100 µg/mL) showed antitumor activity against Ehrlich ascites carcinoma cells [61]. Yu et al. [62] demonstrated that palmitic acid exerted anti-gastric cancer effects by inhibiting cell proliferation and invasion and inducing human gastric cancer cell apoptosis. Zhao et al. [63] reported that palmitic acid significantly increased cellular stress and apoptosis and decreased cellular adhesion and invasion in a transgenic mouse model of endometrial cancer. In addition, Sun et al. [64] demonstrated that fatty acid 2-hydroxylation inhibits colorectal cancer cell proliferation, migration, epithelial-to-mesenchymal transition progression, and tumor growth.

Rajavel et al. [65] reported that β-Sitosterol considerably suppressed the proliferation of A549 cells with no damaging effect on healthy lungs. Further, β-Sitosterol treatment triggered apoptosis via ROS-mediated mitochondrial dysregulation, as evidenced by caspase-3 and 9 activation, Annexin-V/PI positive cells, PARP inactivation, loss of MMP, Bcl-2-Bax ratio alteration, and cytochrome c release. Indeed, in A549 cells, β-Sitosterol administration elevated p53 expression and its phosphorylation at Ser15, while pifithrin-α silenced p53 expression and decreased β-Sitosterol-induced apoptosis. Additionally, NCI-H460 cells (p53 wild) showed β-Sitosterol-induced apoptosis, whereas NCI-H23 cells (p53 mutant) did not. In A549 and NCI-H460 cells, down-regulation of Trx/Trx1 reductase aided in the production of ROS brought on by β-Sitosterol and the cell death caused by mitochondrial apoptosis. Pro-apoptotic, anti-proliferative, anti-metastatic, and anti-invasive effects on tumor cells are some of the anticancer mechanisms of β-sitosterol [66]. By deactivating the TGF-β/Smad2/3/c-Myc pathway, β-sitosterol reduced the autophagy flux and viability of A549 cells in non-small-cell lung cancer both in vitro and in vivo. Supplementing with β-sitosterol in vitro dramatically reduced the ability of lung metastasis cells to bind to fibronectin and laminin. Likewise, β-sitosterol markedly inhibited the growth of transplantable tumors into the lung parenchyma and markedly increased the activity of cytotoxic T-cells and macrophages in melanoma-bearing animals. These immune-boosting effects of β-sitosterol extraordinarily inhibited lung metastasis of transplanted melanoma. It has been demonstrated that β-sitosterol effectively targets excessive neutrophil recruitment in the mouse model of lung chronic infection, promoting the reduction in inflammation [67].

Alkaloids, particularly the isoquinoline alkaloid class, which is abundant in our extract like Berberine, have an anti-lung cancer effect [68]. According to in vitro and in vivo models, isoquinoline alkaloids significantly reduce cancer by causing cell death by autophagy, apoptosis, and cell cycle arrest. The isoquinoline alkaloids exhibited anticancer effects on the A549 and H1299 cell lines, leading to increased protein levels of cleaved PARP and cleaved caspase 3 [69]. By controlling the signaling pathways for the epidermal growth factor receptor/AMP-activated protein kinase (EGFR/AMPK), isoquinoline prevented the growth of non-small cell lung cancer [70]. Also, Berberine-induced caspase 3, 8, and 9-mediated apoptosis in A549 and H1299 xenograft mouse models and triple-negative breast cancer cells [71]. Berberine treatment of A549 cells showed an indication of apoptosis with increased phosphorylation of p38-MAPK and induced protein expression of p53 and FOXO3a. Berberine affected PKC, glycogen synthase kinase 3 beta (GSK-3β), ERK activity, and (NSAID) activated gene-1 (NAG-1) expression, resulting in apoptosis in HCT-116 cells. Berberine leads to G1 cell cycle arrest with the induction of NAG1 and activating transcription factor 3 (ATF3) expression on HCT116 cells [68]. Berberine inhibited cyclooxygenase-2 transcriptional and induced apoptosis [68]. Berberine’s anticancer effects have been associated with DNA and histone modifications [68]. Berberine also repressed HDAC activity and triggered sub-G0/G1 cell cycle arrest in A549 cells [72]. Magnoflorine, which belongs to aporphine alkaloids, recorded antitumor activity. Magnoflorine can inhibit cell proliferation and migration and cause apoptosis. It has been shown that magnoflorine in acute lung injury by lipopolysaccharides (LPS) inhibits the NF-kB [73].

The second is how *Purslane* extract treated lung cancer. Pet. ether extract has anti-lung cancer effects by affecting many biological processes involved in carcinogenesis in general and lung cancer in particular. The extract inhibited inflammation, proliferation, oxidative stress, and angiogenesis and encouraged apoptosis. In the next section, we discussed the relationship between these processes and carcinogenesis, as well as the effect of the *Purslane* extract on them.

*Purslane* extract suppressed proliferation, significantly reducing the Ki67 level (20.42%) compared to the NNK control. Cancer arises as an accumulation of mutations in genes that govern the apoptosis or proliferation process, which results in uncontrolled cell growth. Cancer cells develop a number of skills that are absent from the majority of healthy cells. They develop resistance to growth inhibition, multiply limitlessly, proliferate without relying on growth factors, avoid apoptosis, and invade, spread, and promote angiogenesis [74]. The cellular cycle and proliferation marker Ki-67 antigen is typically employed to estimate the population proliferation of cells and to show the cell growth ratio [30]. This protein is useful in tumoral histopathology because the proportion of Ki-67-positive cells in malignant tumors can be connected to the tumor’s aggressiveness or development parameters. The histological tumor subtype, tumor cellularity, and degree of differentiation, along with other proliferation immunohistochemical markers, such as p53 cellular tumor antigen, are all correlated with the Ki-67 score in patients with lung cancer [75].

*Purslane* extract inhibited angiogenesis, where it significantly reduced MMP9 expression (79%) and IGF-1 62.0%) compared to the NNK control. Since solid tumors need blood flow to grow larger than a few millimeters, cancer cells use angiogenesis processes to produce new blood vessels. Vascular endothelial growth factor is one potent angiogenic agent (VEGF) [76]. MMPs may cleave molecules on the cell surface, such as the tumor suppressor E-cadherin, and can break down the extracellular matrix and basement membranes. MMP9 is a critical enzyme in tumor development and angiogenesis, and it can release growth factors such as insulin development factor (IGF) [77]. Additionally, studies have shown that MMP9 plays a crucial role in the release of VEGF, a major angiogenic factor, from the extracellular matrix, which promotes angiogenesis and tumor growth [78]. Endostatin and other anti-angiogenic agents can create type XVIII collagen, MMP2, and MMP9, for instance, and reveal a hidden epitope inside type IV collagen that induces angiogenesis [79]. Furthermore, TNF-α, IL-8, and other identified pro-angiogenic factors promote the synthesis of MMP-2, -8, and -9 in endothelial cells and control the angiogenesis process [80]. Atherosclerosis and normal wound healing both involve angiogenesis, which is controlled by a variety of growth factors, including IGF-1. IGF-1 encourages rat aorta angiogenesis in vitro and enhances vascular EC migration and tube formation. Retinal angiogenesis requires IGF-1, and by blocking vascular endothelial growth factor signaling, an IGF-1R antagonist prevents retinal neovascularization in vivo [81]. An essential tyrosine kinase receptor, IGF-1R, has a role in angiogenesis, apoptosis, metabolism, differentiation, and cell proliferation. Lung cancer patients often have abnormal IGF-1R activation, which leads to malignant transformation and a bad prognosis for these individuals. Breast cancer, endometrial adenocarcinoma, hepatic cancer, and colorectal cancer have all been shown to have IGF-1R signaling-induced VEGF production [82].

*Purslane* extract induced apoptosis, where it significantly reduced FOXO1 expression (by around 262%) compared to the NNK control. The tumor-negative regulatory transcription factor FOXO1 is abundantly expressed in the lung, heart, and liver, among other human tissues [83]. It can contribute to apoptosis, cell cycle arrest, and DNA damage repair by controlling target gene expression, suppressing tumor cell proliferation, and playing a significant influence in the occurrence, progression, and prognosis of tumors. Low levels of FOXO1 can stimulate the growth of non-small cell lung cancer cells, have a significant inhibitory effect on the cell cycle of lung cancer cells, and trigger apoptosis [84]. FOXO1 is a good target for enhanced P13K/AKT signaling. FOXO1 administers its function by interaction and regulation of several kinds of signaling pathways and axes [85]. Many genes are phosphorylated when PI3K is activated because AKT is translocated to the cytoplasm and nucleus. The phosphorylation of target proteins by Akt can be stimulatory or inhibitory, hence altering or increasing their activity. Akt-induced phosphorylation suppresses FOXO1, reversing its tumor suppressor effect; therefore, activating the P13K/AKT axis might lead to a cancerous state [86]. Furthermore, the PI3K/AKT/FOXO1 signaling pathway has received more attention in recent years because of its importance in lung cancer. Studies have also identified a number of upstream regulators that modulate this axis. One such regulator is the Sema domain of semaphorin 4B (SEMA4B), a novel subtype of semaphorin protein that plays a critical role in the carcinogenesis of non-small cell lung cancer (NSCLC) [87] and also increases nuclear versus cytoplasmic FOXO1 levels through suppression of the PI3K/AKT signaling pathway. This action leads to the prevention of tumor cell development through the binding of FOXO1 to the promoter of cell-cycle-inhibitor p21 that suppresses the growth of tumor cells [88]. In addition, apoptotic genes such as legumain, Fas ligand (FasL), and Bim are upregulated when FOXO1 is activated [89].

*Purslane* extract recorded an anti-inflammatory effect, where it significantly down-regulated FOXO1 expression (62.0%) compared to the NNK control. The IGF-1 signaling participated in almost every stage of lung cancer. For instance, the production of IGF-1 is higher in severe bronchial dysplasia than in benign bronchial epithelial cells, which subsequently interacts with tobacco carcinogens to enhance lung carcinogenesis, suggesting an early function for IGF-1 in the formation of lung malignancies. Additionally, by binding the IGF-1 receptor, exogenous IGF-1 was discovered to stimulate the up-regulation of epithelial–mesenchymal transition (EMT) and to enhance proliferation, invasiveness, metastasis, and ultimately resistance to epidermal growth factor receptor tyrosine kinase inhibitors (EGFR-TKIs) [90]. According to Li et al. [91], IGF1R signaling facilitates inflammation as TNF-induced activation of NF-kB, a critical pathway implicated in inflammation. Similarly, IGF1R is crucial for the start of the inflammatory process since, in mice, blocking the macrophage IGF1/IGF1R signaling axis reduces the NLRP3 inflammasome, a protein complex that is activated in the lung after BLM-induced damage [92]. In the process of inflammation, M2 macrophages promote tissue repair and inflammation resolution upon activation of anti-inflammatory cytokines (e.g., IL13 and CSF1), whereas M1 macrophages contribute to tissue damage following excessive production of pro-inflammatory mediators (e.g., TNF and IL1β) [93]. In this case, the pulmonary milieu enriched in M2 macrophages would be encouraged by the reduced expression of TNF, IL1β, and IL6, as well as the increased levels of CSF1, IL13, and Cd209a observed in CreERT2 lungs. Notably, it has been observed that IL13 guards against acute hypertoxic lung damage, and Cd209a expression was elevated in resolution-phase macrophages following the development of peritonitis in mice [94]. All of these findings are consistent with the notion that IGF1R deficiency in the lung tumor microenvironment led to decreased tumor initiation and progression because it diminished fibrosis, vascularization, and proliferation. It also made it easier to dampen innate and adaptive immunity and resolution of inflammation [95].

*Purslane* extract has antioxidant properties. Compared to the NNK control, it decreased oxidative stress biomarkers (MDA) and increased antioxidant biomarkers (GSH, GR, GST, GPx, CAT, and SOD). High oxygen pressure and elevated endogenous and external oxidative stress are directly experienced by lung tissue. The production of ROS under these circumstances is crucial for the start and advancement of neoplastic development. Numerous mechanisms, including a complex system of antioxidant enzymes, SOD, GPx, GR, and CAT, as well as nonenzymatic antioxidants, GSH, vitamins A, C, D, and E, and β-carotene, protect the lungs against these oxidants. Although antioxidant enzymes can stop the harmful chain of events that ROS start, their incapacity to offset intracellular ROS levels results in metabolic disruptions and cell death. SOD, which catalyzes the dismutation of superoxide anion (O_2_•−) into O_2_ and hydrogen peroxide (H_2_O_2_), is the first line of defense against ROS. CAT or GPx further reduces the H_2_O_2_ produced as a result of O_2_•− being dismutated by SOD to H_2_O. By lowering hydroperoxides, squelching free radicals, and detoxifying xenobiotics, GSH provides defense against oxidative stress. GSH levels in the human lung’s epithelial lining are approximately 140 times greater than those in the bloodstream. GSH and its associated enzymes, including GST, GPx, and GR, make up the GSH-dependent antioxidant system. Protein kinase B, protein phosphatases 1 and 2A, calcineurin, nuclear factor κB, c-Jun N-terminal kinase, apoptosis signal-regulated kinase 1, and mitogen-activated protein kinase are among the signaling pathways that are activated when the GSH/GSSG ratio shifts towards the oxidized state. This decreases cell proliferation and increases apoptosis [96,97]. The *Purslane* Pet. ether extract in the current investigation dramatically decreased oxidative stress parameters (MDA) and enhanced antioxidant parameters.

In the end, we propounded that ether extract of *Purslane* leaves struggles to chemically induce lung cancer in rats via (i) reducing proliferation, (ii) reducing angiogenesis, (iii) increasing apoptosis, (iv) suppressing inflammation, and (v) activating the antioxidant.

## 3. Materials and Methods

### 3.1. Chemicals

4-(Methylnitrosamino)-1-(3-pyridyl)-1-butanone (NNK) was purchased from Santa Cruz Biotechnology Company, Germany. All chemicals and solvents used are analytical grades. Kits of antioxidant parameters including glutathione (GSH), glutathione reductase (GR), glutathione-S-transferase (GST), glutathione peroxidase (GPx), catalase (CAT), superoxide dismutase (SOD) and oxidative stress biomarkers, including MDA, as well as safety profile kits, were acquired from Egypt’s Bio Diagnostic, 29 El-Tahrir St., Dokki, Giza, Egypt. ELISA kits for ICAM (Cat No.:000) and Ki67 (Cat No.:000) were purchased from Sunlong Biotech Co., Ltd., Hangzhou, China.

### 3.2. Authentication of the Plant and Extraction

*Purslane* seeds were identified at the herbarium of Flora and Phytotaxonomy Research, Horticulture Research Institute, Agriculture Research Center, Dokki, Giza, Egypt, to be used as plant material. The voucher specimen (No. 136) was deposited in the herbarium of the Agricultural Research Center (CAIM). Seeds were cultivated in El-Sharkia Governorate, Egypt, during the season of 2022. Leaves were air-dried while they completed dryness in the oven at 40 °C.

*Purslane* leaf powder (2000 g) was exhaustively extracted with petroleum ether (40:60) several times (3 times each, 5 L every week for three weeks) in a lab setting (25 °C) by soaking. A rotary evaporator was used to concentrate the extract (under reduced pressure within a temperature of 35 °C), which was lyophilized and turned into a powder (125 g) and kept at 20 °C until use.

### 3.3. The Study on the A549 Lung Cancer Cell Line (In Vitro Investigation)

After being obtained from the American Type Culture Collection (ATCC), the A549 lung cancer cell lines were maintained under the appropriate preservation conditions. In DMEM (Dulbecco’s modified Eagle’s Medium) supplemented with 10% FBS, 100 U/mL penicillin, and 100 μg/mL streptomycin sulfate, A549 cells were cultivated at 37 °C in humidified conditions with 5% CO_2_. For passaging, the cells were digested using 0.025% trypsin-EDTA. For this experiment, cells in the logarithmic growth phase were used. The in vitro cytotoxicity activity of *Purslane* Pet. ether extract was evaluated using a neutral red uptake test. The tested extract with various doses (25, 50, 100, 200 µg/mL) was added to the 96-well plate culture to maintain the cell density of 10^4^ cells per well for 48 h, and a neutral red uptake test was performed in accordance with Repetto et al. [98]. A relationship between the utilized log concentrations and the neutral red intensity value was used to find out the IC_50_ (the half-maximal inhibitory concentration) of the studied extract. In place of the extract, the medium was administered to the untreated cells (negative control). The positive control, doxorubicin (Mr. = 543.5), showed 100% inhibition. After dissolving the extract in dimethyl sulfoxide (DMSO), the final concentration on the cells was limited to a maximum of 0.2%. Each test was run through at least three times.

After dissolving the extract in dimethyl sulfoxide (DMSO), the final concentration on the cells was limited to no more than 0.2%. Every test was conducted three times or more.

### 3.4. The Study on the Experimental Animals (In Vivo Investigation)

#### 3.4.1. Determination of LD_50_

LD_50_ of the *Purslane* leaves Pet. ether extract was performed [99] as per OECD guideline 425 (OECD 2008) for acute oral toxicity using the Up-and-Down Procedure (UDP) in mice. Extract dosages varied between 0.5 and 10 g/kg body weight. Mice (5 per group) were force-fed the extract orally via a stomach tube, whereas control mice were force-fed saline. For 48 h, the animals were kept under observation and monitored for changes and mortality. For two weeks, the mice that were reminded were monitored. The LD_50_ was calculated by counting the number of dead animals in each concentration over the first 48 h using the BioStat program (BioStat 2009 Build 5.8.4.3 # 2021 analyst Soft Inc., Alexandria, VA, USA). LD_50_ of the *Purslane* Pet. ether extract was 10 g/kg body weight. A dose of 500 mg/kg of *purslane* extract, or 1/20 of the LD_50_, was employed in this investigation.

#### 3.4.2. Animals and Accommodations

The National Research Center’s animal house in Dokki, Cairo, Egypt, provided male albino rats of the Sprague Dawley strains that weighed 180–200 g at 10 weeks of age. These rats were kept in specially designed plastic cages. The following laboratory animal conditions were used to house the animals [100] (20–25 °C, 55–65% humidity, and 10–12 h light/dark cycle). Over 10 months, food and water were freely available.

#### 3.4.3. Lung Cancer Induction

For the purpose of inducing lung cancer, fifty mature male Sprague-Dawley rats weighing 180–200 g were used. For 20 weeks, rats in this group received subcutaneous treatment with 1.5 mg/kg body weight NNK three times per week [101]. At the end of 20 weeks, lung cancer was confirmed histologically using five rats.

#### 3.4.4. Experimental Layout

The four groups of Sprague-Dawley male adult rats were as follows:

Group 1: Using a gastric tube, 20 healthy rats were given 1 milliliter of distilled water orally for 40 weeks during the trial. This group was maintained as a ve− control group.

Group 2: *Purslane* ether extract (500 mg/kg/day) was given orally to 20 healthy rats for 20 weeks (treatment period) after they had been given 1 mL of distilled water orally for 20 weeks (induction phase). As ve+ of ether extract, this group was retained.

Group 3: For 20 weeks, twenty healthy rats got three weekly subcutaneous injections of 1.5 mg/kg body weight NNK. Following this, they were given 1 mL of distilled water orally. This group was retained as a control group for cancer.

Group 4: For 20 weeks, 20 healthy rats got three weekly subcutaneous injections of 1.5 mg/kg body weight NNK. Following this, they were given 500 mg/kg/day of *Purslane* ether extract orally for the duration of the treatment period. This group continued to be treated.

Rats were fasted overnight after 40 weeks. A 0.2 mL/100 g body weight dose of ketamine/xylazine (87 and 13 mg/kg) dissolved in normal saline was administered to the fasted rats to induce anesthesia [102]. The abdominal aorta was used to obtain blood samples. Using a Sigma Laborzentrifugen (Osterode am Harz, Germany), each blood sample was centrifuged at 3500× *g* for 10 min. The serum was then separated and kept at −20 °C until further examination. Vital organs were taken out, weighed, and cleaned with ice-cold saline after permission. For histopathologic examination, the right kidney, liver, and lung were subsequently submerged in 10% buffered formaldehyde. In order to do subsequent chemical experiments, the left lung was kept at −80 °C.

### 3.5. Biochemical Assessments

Liver function: total protein, albumin, aspartate aminotransferase (AST), and alanine aminotransferase (ALT) [103,104,105] were determined in serum spectrophotometrically—the difference between total protein and albumin calculated Globulin [106]. Kidney functions: urea and uric acid [107,108] were estimated in serum spectrophotometrically.

Reduced glutathione (GSH), glutathione reductase (GR), glutathione-S-transferase (GST), glutathione peroxidase (GPx), catalase (CAT), and superoxide dismutase (SOD) were measured as antioxidant indicators in lung homogenates according to Griffith [109], Goldberg and Spooner [110], Paglia and Valentine [111], Habig et al. [112], Beers and Sizer [113], and Fridovich [114], respectively.

Oxidative stress biomarker MDA was determined spectrophotometrically in lung homogenates according to Ohkawa et al. [115].

According to the manufacturer’s instructions, using an enzyme-linked immunosorbent assay, serum ICAM, Ki67, MPO, and collagen were determined using ELISA kits.

### 3.6. Quantitative Real-Time PCR

Use the RNAeasymini Kit (Qiagen, Germany) (Cat. No./ID: 74104) to extract RNA from rat lung tissues. Next, the concentration and purity of the extracted total RNA were assessed using the NanoDrop one microvolume UV spectrophotometer (Thermo Fisher Scientific, Waltham, MA, USA). Use the Revert Aid First Strand cDNA Synthesis Kit (Thermo Fisher Scientific, Waltham, MA, USA) (Cat. No.: K1621) as stated by the manufacturer to transform RNA from each treatment into first-strand cDNA. Table 2 contains the primer sequences for the examined genes. Maxima SYBR Green qPCR Master Mix (2X) (Thermo Fisher Scientific, Waltham, MA, USA) (Cat. No.: K0221) was used to normalize the *MMP9*, *IGF1*, and *FOXO1* gene expression levels with respect to the *β-actin* transcript. Following the steps described by Livak and Schmittgen [116], the expression levels were subsequently calculated using the 2^−ΔΔCT^ technique. The following reaction conditions were used for 40 cycles of amplification: 95 °C for 10 min, 95 °C for 15 s, 55 °C for 30 s, and 72 °C for 30 s. Using the DNA Technology Detecting Thermocycler DT Lite 4S1, gene expression was quantified.

### 3.7. Histology Assay

After being harvested, the liver, kidneys, and lungs were fixed in 10% buffered formalin, dehydrated, cleaned with xylene, and then embedded in paraffin. In order to prevent potential bias, thick slices (4–5 μm) were prepared, stained with Hematoxylin and Eosin (H&E) [117], and examined by a skilled pathologist using a light microscope (Leica UA510CA, Leica Microsystems, Wetzlar, Germany). The specimen’s identity was maintained confidential during image capture and analysis. The grades for histopathological changes were (1), (2), and (3), which denoted mild, moderate, and severe changes, respectively, and (0), which indicated no changes.

### 3.8. Qualitative Analysis of the Chemical Composition in Leaves from Pet. Ether Extract of Purslane by HPLC/HR-QTOF-MS/MS

Liquid chromatography–mass spectrometry analysis was used to identify the phytoconstituents of *Purslane* [30,44,45]. HPLC–MS/MS analysis was carried out in the Proteomics and Metabolomics Research Program of the Basic Research Department at the Children’s Cancer Hospital, Cairo, Egypt.

#### 3.8.1. Sample Preparation

The ethereal extract of *Purslane* was prepared and freeze-dried, and an aliquot of five milligrams was taken in one thousand microliters of the solvent prepared from (H2O:CH_3_OH:ACN, 50:25:25) *v*/*v*. It was mixed by vortexing for two minutes and then ten minutes using ultrasonication at a frequency of 30 kHz; the sample was completely dissolved in the basic solution. Then, a portion of 20 microliters of the basic solution was diluted with 1000 microliters of the mixture (H_2_O:MeOH:ACN, 50:25:25 *v*/*v*) and centrifuged for 10 min at a speed of 10,000 rpm. Finally, 10 microliters of the basic solution were injected at a dose of 2.5 micrograms/microliter. A blank sample containing about 10 microliters of the prepared solvent was injected. Then, both positive and negative injection modes were used for the sample.

#### 3.8.2. Instruments and Acquisition Method

For the mass spectrometry (MS) experiment, a triple TOF 5600+ system with a duo-spray source operating in the ESI mode (ABSCIEX, Concord, ON, Canada) was utilized. The sprayer capillary and de-clustering potential voltages were −4500 and −80 V, respectively, in the (+ve) and (−ve) modes. The ion source temperature was fixed at 600 °C, the curtain gas was at 25 pressures, and ion source gases 1 and 2 were at 40 psi. A collision energy (CE) 35 V of negative mode, a CE spreading of 20 V, and an ion tolerance of 10 ppm were used. The TripleTOF5600+ was conducted using an approach known as information-dependent acquisition (IDA). Batches for collecting MS and MS/MS data were built using Analyst-TF 1.7.1 software (AB Sciex, Foster City, CA, USA). MS and MS/MS data full-scan were gathered using the IDA approach. The operation conditions include ammonium format (HCOONH_4_) buffer pH = 8 with 1% methanol and 100% acetonitrile (ACN), in-line filter disks, and DI water plus 0.1% formic acid (HCOOH) were the mobile phases. The pre-column (0.5 µm × 3.0 mm, Phenomenex, Torrance, CA, USA), column temperature, and Xbridge C18 column (3.5 µm, 2.1 × 50 mm, waters) were maintained at 40 °C with a flow rate of 0.3 mL/min.

#### 3.8.3. LC–MS/MS Data Processing

The material was subjected to a thorough small-molecule analysis using open-source software without any targeting. Depending on the acquisition mode, respect positive (2737) MS-DIAL 4.8 and negative (1573 records) databases were utilized as reference databases. Using the total ion chromatogram (TIC), the MSDIAL output was utilized to run PeakView 2.2 once more using the Master View 1.1 software (AB SCIEX, Toronto, ON, Canada) for feature (peaks) validation based on the criteria. Features that were aligned had sample intensities larger than three and signal-to-noise ratios greater than five.

### 3.9. Statistical Analysis

The findings were displayed as the mean ± SEM. The Sigma Plot Ver. 125 is used to calculate the data. The statistical significance between the control and the studied extracts was assessed using the Student *t*-test. A one-way analysis of variances (one-way ANOVA) was used to determine whether there was a statistically significant difference between the groups, and the Tukey–Kramer multiple comparison test was performed. GraphPad Prism version 8 (GraphPad Software Inc., San Diego, CA, USA) was utilized for data analysis. For every test, a significant difference was defined as *p* < 0.05.

## 4. Conclusions

Tobacco consumption is by far the most significant risk factor for lung cancer worldwide. The current study concluded that the administration of *Purslane* leaf Pet. ether extracts at a dose of 500 mg/kg/day orally for 20 weeks exhibited anticancer effects on A540 cell lines and NNK-induced lung cancer rats, where ICAM, the lung cancer biomarker, was significantly reduced. *Purslane* inhibited proliferation, angiogenesis, and inflammation and encouraged apoptotic and antioxidant systems in comparison to cancer control. The histopathological examination of the lungs confirmed the previous results. *Purslane*’s anti-lung cancer effect may be associated with the active components that were identified by LC-Mass. This work may result in clinical trials following a reexamination of this sort of small experimental animal, its progeny, and an experiment involving a larger animal, as the extract has a good safety margin.

## Data Availability

Samples of the compounds are not available from the authors.

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
