# Peer review of "The Phytochemical Profile of the Petroleum Ether Extract of Purslane Leaves and Its Anticancer Effect on 4-(Methylnitrosamino)-1-(3-pyridyl)-1-buta-4 None (NNK)-Induced Lung Cancer in Rats"

_ijms, 2024, doi:10.3390/ijms252313024_

Round 1
Reviewer 1 Report
Comments and Suggestions for Authors
The article shows the phytochemical characterization of Purslane petroleum ether extract and its potential in combating lung cancer induced by 4-(methylnitrosamino)-1-(3-pyridyl)-1-butane-4-nonone (NNK) in rats. Despite having citations from 2024, 38% of the references listed are out of date and are more than 10 years old. The article may be published after the authors respond to the comments.
1. Reduce similarity in text to < 20% (Percentage match: 31% - iThenticate report)
2. Language corrections (it is recommended that the manuscript be reviewed by a native English speaker).
3. Abstract line 39- Modify identified by characterized.
4. The extract evaluated was petroleum ether, which is a nonpolar extract. The most appropriate method for analyzing nonpolar extracts and characterizing their compounds is GC-MS. Why did the authors use HPLC-QTOF/HR-MS/MS for the analysis? My suggestion is that the analysis be done by GC-MS, which is the most appropriate technique for this type of extract.
5. Introduction line 63- Format the sentence “Purslane (S.N: Portulaca oleracea L., Family: Portulaceae) is known in Egypt as the Regla plant” with the appropriate font size.
6. Results and discussion
a) lines 113-114- Format the sentence with the appropriate font size.
b) lines 125-129- Format the sentence with the appropriate font size.
c) Number the results and discussion subheadings sequentially.
d) Line 184- sum?? would not be mean??
e) Figure 4: it would be better to write only letters or symbols to differentiate the statistical analysis and not mix them to make it more standardized.
f) In Figure 7 it is not clear which photomicrographs represent the negative control and Purslane-treated groups.
h) In Figure 8, what magnification was used in photomicrographs G and H?
i) In Figure 9, what magnification was used in photomicrographs G and H?
j) Why did the authors use HPLC-QTOF/HR-MS/MS for the analysis and not GC-MS?
k) Table 1. Compound 4: 320.239 0[M+-H2O]+ would not be 320.2390 [M+H-H2O]+? Compound 5: would not be 324.0550 [M+H-H2O]+?
Compound 30: Would a loss of 17u be [M+-H2O]+ or [M+H-H2O]+?
Compounds 48 and 49, m/z 309.2895 [M+H-H2O-60(4CH3)-2H]+ and 309.2600 [M+H-H2O-60(4CH3)]+, respectively, is correct? See also lines 599 and 601 on page 28.
l) Check and standardize the fragmentations for the alkaloids
Would a loss of 17u be [M+-H2O]+ or [M+H-H2O]+?
Would a loss of 18u be [M+-H2O]+ or [M+H-H2O]+?
m) What does "*, **, and ***” mean in table 1? Add in caption.
n) m/z is in italic.
o) In fragmentations several indexes are not subscribed. Example [M+-(CH3)2NH-CH3OH] and not [M+-(CH3)2NH-CH3OH]. Make these corrections throughout the manuscript.
p) Scheme 1. m/z 342 – 2OH = m/z 307 loss of 35 amu? Wouldn't it be m/z 308? Check the values in scheme 1.
q) Scheme 2. m/z 206 – CO = m/z 179 loss of 27 amu? Wouldn't it be m/z 178? Check the values in scheme 2. m/z 356 – H2O = m/z 339 loss of 17 amu?
r) line 520 - Formate the sentence with the appropriate font size
s) Scheme 4. m/z 328 – H2O = m/z 311 loss of 17 amu? Check the values in scheme 4. m/z 311 – OCH3 = m/z 279 loss of 32 amu?
t) Scheme 5. m/z 344 – H2O = m/z 327 loss of 17 amu? m/z 327 – H2O = m/z 310 loss of 17 amu? Check the values in scheme 5.
u) Scheme 6. Compound 4 m/z 135 – H2O = m/z 122 loss of 13 amu? Compound 2 - m/z 191 + m/z 107 = m/z 298 not m/z 300? Check the values in scheme 6.
v) line 605 – Compound 47 m/z 369.2416 [M+H- H2O-H-CH3]- loss m/z 16 not m/z 355.1109 [M+H-H2O-2H-2CH3]-, referring to the loss of -H-CH3.
w) The same occurs for compound 53 in line 607, m/z 363.2307 [M+H- H2O-H-CH3]- loss m/z 16 not m/z 349.5022 [M+H-H2O-2H-2CH3]-, referring to the loss of -H-CH3.
x) lines 632-636 not in bold.
y) lines 644-647 standardize compound names by starting with either an uppercase or lowercase letter, but do not mix the two.
7. Materials and methods
a) Lines 884- 886
was exhaustively extracted with ether (40:60)- clarify 40:60 of which solventes?
under laboratory conditions by soaking. What would be the conditions by soaking and for what time?
The extract was concentered using a Rotary Evaporator - conditions by soaking?
b) Line 889 - A549 lung cancer cell (ATCC???)
c) Lines 954-957 -Formate the sentence with the appropriate font size
d) Detail the analysis conditions of HPLC/HR-QTOF-MS/MS
e) Line 1011 Gas 1 and gas 2????
8. References -38% of the listed references are outdated; please update them to not older than 2014.
Comments on the Quality of English LanguageLanguage corrections (it is recommended that the manuscript be reviewed by a native English speaker).
Author Response
Author’s Response
Responses to reviewers’ comments for “The phytochemical profile and unveiling of the promising potential of Purslane petroleum ether extract in combating lung cancer induced by 4-(Methylnitrosamino)-1-(3-pyridyl)-1-butanone (NNK) in rats.” (Manuscript ID: ijms-3271464).
Dear Editor,
We thank the reviewers for their insightful comments, which enabled us to improve the manuscript. We hope that you will accept it after revision. According to the reviewers' suggestions, the manuscript has been revised carefully. The detailed corrections raised by the reviewers have been addressed point by point and highlighted in blue in the revised manuscript.
Reviewer 1:
Comments and Suggestions for Authors
Despite having citations from 2024, 38% of the references listed are out of date and are more than 10 years old. The article may be published after the authors respond to the comments.
Response: Dear Reviewer, we have 34 references out of 10 years ago, 20 references in materials and methods; we can not change them.
We changed 21 references into new ones
- 5- Damavandi, R. D.; Shidfar, F.; Najafi, M.; Janani, L.; Masoodi, M.; Heshmati, J.; Ziaei, S. Effect of portulaca oleracea (purslane) extract on inflammatory factors in nonalcoholic fatty liver disease: A randomized, double-blind clinical trial. Journal of Functional Foods 2023, 102, 105465. Instead of “5-Chan, K.; Islam, M. W.; Kamil, M.; Radhakrishnan, R.; Zakaria, M. N.; Habibullah, M.; Attas, A . The analgesic and anti-inflammatory effects of Portulaca oleracea subsp. sativa (Haw.) Celak. J. Ethnopharmac.2000, 73, 3, 445–451.doi: 10.1016/s0378-8741(00)00318-4.
- 7- Khazdair, M. R.; Anaeigoudari, A.; Kianmehr, M. Anti-Asthmatic Effects of Portulaca Oleracea and its Constituents, a Review. Journal of Pharmacopuncture 2019;22[3]:122-130 doi.org/10.3831/KPI.2019.22.01. instead of Dkhil, M. A.;Moniem, A. E. A.; “7- Al-Quraishy, S.; Saleh, R. A. Antioxidant effect of purslane (Portulaca oleracea) and its mechanism of action. Med. Plants Res.2011, 5(9), 1589–1593.”
- 8- de Souza,P. G.; Rosenthal,A.; Ayres, E. M. M.; Teodoro, A. J. Potential Functional Food Products and Molecular Mechanisms of Portulaca Oleracea L. on Anticancer Activity: A Review. Oxidative Medicine and Cellular Longevity 2022, 2022, Article ID 7235412, 9 pages. doi.org/10.1155/2022/7235412. Instead of “Hozayen, W.; Bastawy, M.; Elshafeey, H. Effects of aqueous purslane (Portulaca Oleracea) extract and fish oil on gentamicin nephrotoxicity in albino rats.Nature Sci.2011, 9(2), 47–62.”
- 9- Obukohwo, O. M. Nutraceutical health benefit and safety utility of Portulaca oleracea: A review focus on neuroendocrine function. Clinical Traditional Medicine and Pharmacology 2024, 5, 200168. https://doi.org/10.1016/j.ctmp.2024.200168. Instead of “9- Wang, C. Q.; Yang, G. Q. Betacyanins from Portulaca oleracea ameliorate cognition deficits and attenuate oxidative damage induced by D-galactose in the brains of senescent mice. Phytomed. 2010, 17(7), 527–532.”
- 12- Salman, K. H.; Mahmoud, E. A.; Abd-Alla, A. A. Preparing Untraditional Kishk formula with Purslane as natural source of bioactive compounds. J. of Food and Dairy Sci., Mansoura Univ. 2020, 11 (11):299-305. DOI: 10.21608/jfds.2020.126744. Instead of “12- Gonnella, M.; Charfeddine, M.; Conversa, G.; Santamaria, P. Purslane: a review of its potential for health and agricultural aspects. J. Plant Sci. Biotech. 2010, 4(1), 131–136.”
- 22- Ghorani, V.; Saadat, S.; Khazdair, M. R.; Gholamnezhad, Z.; El-Seedi, H.; Boskabady, M. H. Phytochemical Characteristics and Anti-Inflammatory, Immunoregulatory, and Antioxidant Effects of Portulaca oleracea L.: A Comprehensive Review. Evidence-Based Complementary and Alternative Medicine 2023, 2023, Article ID 2075444, 29 pages https://doi.org/10.1155/2023/2075444. Instead of 22- Youguo, C.; Zongji, S.; Xiao Ping, C. Evaluation of free radicals scavenging and immunity-modulatory activities of Purslane Int. J. Biol.Macromol.2009, 45(5), 448–452.
- 23- Xia, L.; Yang, M.; Liu, Y. Portulaca oleracea L. polysaccharide inhibits ovarian cancer via inducing ACSL4-dependent ferroptosis. Aging 2024, 16, (6), 5108-5122. Instead of 23- Shen, H.; Tang, G.; Zeng, G.; Yang,Y.; Cai, X.; Li, D.; Lui, H.; Zhou, N. Purification and characterization of an antitumor polysaccharide from Portulaca oleracea Carb. Pol.2013, 93(2), 395–400.doi: 10.1016/j.carbpol.2012.11.107.
- 24- Zhao, R.; Shao, X.; Jia, G.; Huang,Y..; Liu, Z.; Song, B.; Hou, J.Anti-cervical carcinoma effect of Portulaca oleracea polysaccharides by oral administration on intestinal dendritic cells. BMC Complementary and Alternative Medicine 2019, 19:161. doi.org/10.1186/s12906-019-2582-9. Instead of 24- Zhao, R.; Gao, X.; Cai, Y.; Shao, X.; Jia, G.; Huang, Y.; Qin, X.; Wang, J.; Zheng, X. Antitumor activity of Portulaca oleracea L. polysaccharides against cervical carcinoma in vitro and in vivo. Car. Pol.2013, 96(2), 376–383.doi: 10.1016/j.carbpol.2013.04.023.
- Farag, M-A.; Shakour, Z-T. Metabolomics driven analysis of 11 Portulaca leaf taxa as analysed via UPLC-ESI-MS/MS and chemometrics. Phytochemistry. 2019,161,117-129. https://doi.org/10.1016/j.phytochem.2019.02.009 instead of “ Yue, M.E.; Jiang, T.F.; Shi, Y.P. Simultaneous determination of noradrenaline and dopamine in Portulaca oleracea L. by capillary zone electrophoresis. J. Sep. Sci. 2005, 28(4), 360-364. https://doi.org/10.1002/jssc.200400045
- Qing, Z.; Xu, Y.; Yu, L.; Liu, J.; Huang, X.; Tang, Z.; Cheng, P.;, Zeng, J. Investigation of fragmentation behaviours of isoquinoline alkaloids by mass spectrometry combined with computational chemistry. Scientific Reports. 2020 , 20;10(1):733. https://doi.org/10.1038/s41598-019-57406-7. instead of “ Zhang, Y.; Shi, Q.; Shi, P.; Zhang, W.; Cheng, Y. Characterization of isoquinoline alkaloids, diterpenoids and steroids in the Chinese herb Jin‐Guo‐Lan (Tinosporasagittata and Tinosporacapillipes) by high‐performance liquid chromatography/electrospray ionization with multistage mass spectrometry. Rapid communications in mass spectrometry 2006,20(15), pp.2328-2342. https://doi.org/10.1002/rcm.2593
- Singh A, Bajpai V, Kumar S, Rawat AK, Kumar B. Analysis of isoquinoline alkaloids from Mahonia leschenaultia and Mahonia napaulensis roots using UHPLC-Orbitrap-MSn and UHPLC-QqQLIT-MS/MS. Journal of pharmaceutical analysis. 2017, 7(2):77-86. https://doi.org/10.1016/j.jpha.2016.10.002. instead of “Stévigny, C.; Jiwan, J.L.H.; Rozenberg, R.; de Hoffmann, E.; Quetin‐Leclercq, J. Key fragmentation patterns of aporphine alkaloids by electrospray ionization with multistage mass spectrometry. Rapid communications in mass spectrometry2004, 18(5), pp.523-528.“
- Xiang QiuYue X.Q.; Hashi Y.; Chen ZiLin C.Z. Simultaneous detection of eight active components in Radix Tinosporae by ultra high performance liquid chromatography coupled with electrospray tandem mass spectrometry. Journal of Separation Science. 2016, 39(11) 2036-2042. instead of “Chia, Y.C.; Chang, F.R.; Li, C.M.; Wu, Y.C. Protoberberine alkaloids from Fissistigma balansae. Phytochem. 1998, 48(2), pp.367-369.”
- 50- Yan, H.; Zhang, S.; Yang, L.; Jiang, M.; Xin, Y.; Liao, X.; Li, Y.; Lu, J. The Antitumor Effects of α-Linolenic Acid. J. Pers. Med. 2024, 14, 260. https://doi.org/10.3390/ jpm14030260. Instead of 50- Yang, L.; Yuan, J.; Liu, L.; Shi, C.; Wang, L.; Tian, F.; Liu, F.; Wang, H.; Shao, C.; Zhang, Q. α-linolenic acid inhibits human renal cell carcinoma cell proliferation through PPAR-γ activation and COX-2 inhibition. Lett. 2013, 6, 197–202.
- 54- Polavarapu, S.; Mani, A. M.; Gundala, N. K. V.; Hari, A. D.; Bathina, S.; Das, U. N. Effect of Polyunsaturated Fatty Acids and Their Metabolites on Bleomycin-Induced Cytotoxic Action on Human Neuroblastoma Cells In Vitro. PLoS ONE 2014, 9(12): e114766. doi:10.1371/journal.pone.0114766. instead of “54- Rose, D. P.; Hatala, M. A.; Connolly, J. M. Rayburn J. Effect of diets containing different levels of linoleic acid on human breast cancer growth and lung metastasis in nude mice. Res. 1993, 53, 4686-4690.”
- 57- Vaezi, M. A.; Safizadeh, B.; Eghtedari, A. R.; Ghorbanhosseini, S. S.; Rastegar, M.; Salimi, V.; Tavakoli-Yaraki, M. 15-Lipoxygenase and its metabolites in the pathogenesis of breast cancer: A doubleedged sword. Lipids in Health and Disease 2021, 20:169. doi.org/10.1186/s12944-021-01599-2. Instead of 57- Tavakoli-Yaraki, M.; Karami-Tehrani, F. Apoptosis Induced by 13-S-hydroxyoctadecadienoic acid in the Breast Cancer Cell Lines, MCF-7 and MDA-MB-231. Iran J Basic Med Sci. 2013; 16: 653-9.
- 58- Li, M.-Y.; Yuan, H.-L.; Ko, F. W. S.; Wu, B.; Long, X.; Du, J.; Wu, J.; Ng, C. S. H.; Wan, I. Y. P.; Mok, T. S. K.; Hui, D. S. C.; Underwood, M. J.; Chen, G. G. Antineoplastic Effects of 15(S)-Hydroxyeicosatetraenoic Acid and 13-S-Hydroxyoctadecadienoic Acid in Non–Small Cell Lung Cancer. Cancer 2015, 3130-3145. DOI: 10.1002/cncr.29547. Instead of 58- Yuan, H.; Li, M-Y.; Ma, L.T.; Hsin, M.K.Y.; Mok, T.S.K.; Underwood, M.J.; Chen, G.G. 15-Lipoxygenases and its metabolites 15(S)-HETE and 13(S)-HODE in the development of non-small cell lung cancer. Thorax 2010, 65: 321e326. doi:10.1136/thx.2009.122747.
- 74- Yoon, S. W.; Jeong, J. S.; Kim, J. H.; Aggarwal, B. B. Cancer prevention and therapy: integrating traditional Korean medicine into modern cancer care. Integrative Cancer Therapies 2014, Vol. 13(4) 310–331. DOI: 10.1177/1534735413510023. Instead of “74- Gupta, S.C.; Kim, JH; Prasad, S.; Aggarwal, BB. Regulation of survival, proliferation, invasion, angiogenesis, and metastasis of tumor cells through modulation of inflammatory pathways by nutraceuticals. Cancer Metast. Rev. 2010, 29(3), 405–434. doi:10.1007/s10555-010-9235-2.”
- 78- Li, S.; Pritchard, D.M.; Yu, L.-G. Regulation and Function of Matrix Metalloproteinase-13 in Cancer Progression and Metastasis. Cancers 2022, 14, 3263. https:// doi.org/10.3390/cancers14133263. Instead of “78- Lederle, W.; Hartenstein, B.; Meides, A.; Kunzelmann, H.; Werb, Z.; Angel, P.; Mueller, MM. MMP13 as a stromal mediator in controlling persistent angiogenesis in skin carcinoma. 2010, 31(7), 1175-84. doi: 10.1093/carcin/bgp248.”
- 79- Guirado, E.; George, A. Dentine matrix metalloproteinases as potential mediators of dentine regeneration. Eur Cell Mater. 2022, 42: 392–400. doi:10.22203/eCM.v042a24. instead of “Löffek, S.; Schilling, O.; Franzke, C.W. Series "matrix metalloproteinases in lung health and disease": Biological role of matrix metalloproteinases: a critical balance. Eur Respir J. 2011, 38(1), 191-208. doi: 10.1183/09031936.00146510. “
- 81- Macvanin, M.; Gluvic, Z.; Radovanovic, J.; Essack, M.; Gao, X.; Isenovic, E.R. New insights on the cardiovascular effects of IGF-1. Front. Endocrinol. 2023, 14:1142644. doi: 10.3389/fendo.2023.1142644. instead of “81- Delafontaine P, Song YH, Li Y. Expression, regulation, and function of IGF-1, IGF1R, and IGF-1 binding proteins in blood vessels. Arteriosclerosis thrombosis Vasc Biol (2004) 24(3):435–44. doi: 10.1161/01.ATV.0000105902.89459.09”
- 82- Zhou, F.; Nie, L.; Feng, D.; Guo, S.; Luo, R. MicroRNA-379 acts as a tumor suppressor in non-small cell lung cancer by targeting the IGF-1R-mediated AKT and ERK pathways. Oncology Reports 2017, 38, 1857-1866. DOI: 10.3892/or.2017.5835. instead of Yin, M.; Guan, X.; Liao, Z.; Wei, Q. Insulin-like growth factor-1 receptor-targeted therapy for non-small cell lung cancer: a mini-review. Am J Transl Res 2009;1(2):101-114.
Commented 1: 1. Reduce similarity in the text to < 20% (Percentage match: 31% - iThenticate report)
Response 1: The comment is made.
Commented 2: Language corrections (it is recommended that the manuscript be reviewed by a native English speaker).
Response 2: The comment is made.
Commented 3: 3. Abstract line 39- Modify identified by characterized.
Response 2: The comment is made, line 40.
Commented 4: The extract evaluated was petroleum ether, which is a nonpolar extract. The most appropriate method for analyzing nonpolar extracts and characterizing their compounds is GC-MS. Why did the authors use HPLC-QTOF/HR-MS/MS for the analysis? My suggestion is that the analysis be done by GC-MS, which is the most appropriate technique for this type of extract.
Response 4: We thank the reviewer for pointing this out.
The suggestion for the analysis by GC-MS, which is the most appropriate technique for this type of extract; is great suggestion but this is the first study on the ether extract, we wanted to be the first to discover the majority of the phytoconstituents from the various classes that are anti-lung cancer. This is achieved by using high resolution LC-MS/MS and not by GC-MS. HPLC-QTOF/HR-MS/MS is more sensitive for identification of all phytoconstituents; polyphenols; (alkaloids, terbens, flavonoids,…..etc.), fatty acids, sterols and so on unlike GC-MS, which specializes in identifying fatty acids and sterols,..
Commented 5: Introduction line 63- Format the sentence “Purslane (S.N: Portulaca oleracea L., Family: Portulaceae) is known in Egypt as the Regla plant” with the appropriate font size.
Response 5: The comment is made.
Commented 6:
Results and discussion
- lines 113-114- Format the sentencewith the appropriate font size.
Response: The comment is made.
- lines 125-129- Format the sentence with the appropriate font size.
Response: The comment is made.
- Number the results and discussion subheadings sequentially.
Response: The comment is made throughout the manuscript.
- Line 184- sum?? would not be mean??
Response: It means the average of 6 replicates.
- Figure 4: it would be better to write only letters or symbols to differentiate the statistical analysis and not mix them to make it more standardized.
Response: The comment is made.
- In Figure 7 it is not clear which photomicrographs represent the negative control and Purslane-treated groups.
Response: The word respectively was added to show the differences between the two groups.
- In Figure 8, what magnification was used in photomicrographs G and H?
Response: The magnification was added (G X 100 and H X 200).
- In Figure 9, what magnification was used in photomicrographs G and H?
Response: The magnification was added (G X 100 and H X 200).
- Why did the authors use HPLC-QTOF/HR-MS/MS for the analysis and not GC-MS?
Response: We thank the reviewer for pointing this out.
- The analysis HPLC-QTOF/ HR-MS/MS is more sensitive for identification of all phytoconstituents; polyphenols; (alkaloids, terbens, flavonoids,…..etc.), fatty acids, sterols and so on.
- HPLC-HR-MS/MS (high-performance liquid chromatography-high resolution mass spectrometry) is a technique that unifies the application of HPLC with HR-MS. Because of the high sensitivity and good separation ability of HPLC and the sensitivity of HR-MS, this technique has been widely used for structure identification, qualitative determination, and fingerprint analysis.
- k) Table 1. Compound 4: 320.239 0[M+-H2O]+would not be 320.2390 [M+H-H2O]+? Compound 5: would not be 324.0550 [M+H-H2O]+?
Compound 30: Would a loss of 17u be [M+-H2O]+ or [M+H-H2O]+?
Compounds 48 and 49, m/z 309.2895 [M+H-H2O-60(4CH3)-2H]+ and 309.2600 [M+H-H2O-60(4CH3)]+, respectively, is correct? See also lines 599 and 601 on page 28.
Response: We thank the reviewer for pointing this out.
- Table 1. Compound 4: 320.239 0[M+-H2O]+would not be 320.2390 [M+H-H2O]+?
- The number was corrected to 230.2390[M+-H2O]+in the table 1.
- Compound 5: would not be 324.0550 [M+H-H2O]+ ?
- The number was corrected to 324.0550 [M-H2O]+ in the table 1.
- Compound 30: Would a loss of 17u be [M+-H2O]+or [M+H-H2O]+?
Dear reviewer:
- Compound 30 is fatty acid with molecular weight 298. From MS/MS spectral data (please see suppl.file, fig. S3) in negative (-ve) ionization( deprotonation) mode it produced molecular ion peak at m/z 2425 [M-H]- and characteristic fragment at m/z 279.2344[M-H-H2O]- due to loss of water molecules.
- Compounds 48 and 49, m/z 309.2895 [M+H-H2O-60(4CH3)-2H]+and 309.2600 [M+H-H2O-60(4CH3)]+, respectively, is correct? See also lines 599 and 601 on page 28.
- Compounds 48 and 49 were revised in table and text.
- l) Check and standardize the fragmentations for the alkaloids
Would a loss of 17u be [M+-H2O]+ or [M+H-H2O]+?
Would a loss of 18u be [M+-H2O]+ or [M+H-H2O]+?
Response: Dear reviewer:
- We thank you for pointing this out
- The fragmentations for the alkaloids would be [M+-H2O]+in all isoquinoline class (with a positive charge on its nitrogen atom) and [M+H-H2O]+ for class of amino-alkaloids, as Coumaroyltyramine, Caffeoyltyramine, Feruloyltyramine, and Feruloyloctopamine (without a positive charge on its nitrogen atom), terbens, and sterols
- m) What does "*, **, and ***” mean in table 1? Add in caption
Response: We thank the reviewer for pointing this out.
- "*, **, and ***” means the gradation of compound concentrations from lowest to highest (*-***).
- (***); the most highly concentrated compounds in the extract.
- We added the mean of "*, **, and ***”in the caption.
- n) m/z is in italic
Response: We thank the reviewer for pointing this out.
- The comment was done throughout the manuscript.
o). In fragmentations several indexes are not subscribed. Example [M+-(CH3)2NH-CH3OH]
and not [M+-(CH3)2NH-CH3OH]. Make these corrections throughout the manuscript.
Response: We thank the reviewer for pointing this out.
- The correction was made throughout the manuscript.
- p) Scheme 1. m/z342 – 2OH = m/z307 loss of 35 amu? Wouldn't it be m/z 308? Check the values in scheme 1.
Response: We thank the reviewer for pointing this out.
- Scheme 1. Was revised and the correction was made according to spectral MS/MS data in (Suppl. file, fig. S32).
- q) Scheme 2. m/z206 – CO = m/z179 loss of 27 amu? Wouldn't it be m/z 178? Check the values in scheme 2. m/z 356 – H2O = m/z 339 loss of 17 amu?
Response: We so sorry.
- Scheme 2. was revised and the correction was made in green color.
- We hope this addition is more suitable
- r) line 520 - Formate the sentence with the appropriate font size
Response: The comment was done.
- s) Scheme 4. m/z328 – H2O = m/z311 loss of 17 amu? Check the values in scheme 4. m/z 311 – OCH3 = m/z 279 loss of 32 amu?
Response: We so sorry
- Scheme 4. was revised and the correction was made in green color.
- We hope this addition is more suitable
- t) Scheme 5. m/z344 – H2O = m/z327 loss of 17 amu? m/z 327 –
Response: We so sorry
- Scheme 5. was revised and the correction was made in scheme, table and text with green color.
- We hope this addition is more suitable
- u) Scheme 6. Compound 4 m/z135 – H2O = m/z122 loss of 13 amu? Compound 2 - m/z 191 + m/z 107 = m/z 298 not m/z 300? Check the values in scheme 6.
Response: we so sorry and we thank you a lot for your comments.
- Compound 4 m/z135 – H2O= m/z 122 was corrected to135 – NH2= m/z
- Compound 2 - m/z191+ m/z 107 was corrected to m/z 193+107= m/z 300
- We hope this addition is more suitable
- v) line 605 – Compound 47 m/z 2416 [M+H- H2O-H-CH3]-loss m/z16 not m/z 355.1109 [M+H-H2O-2H-2CH3]-, referring to the loss of -H-CH3.
Response: Many thanks; you improved our manuscript a lot.
- line 605 – Compound 47 The m/z 2416 [M+H-H2O-H-CH3]-was corrected to 369.2416[M+H-2H-2CH3]+, and The m/z 355.1109 [M+H-H2O-2H-2CH3]-, was corrected to 355.1109[M+H-H-3CH3]+.
- We hope this addition is more suitable
- w) The same occurs for compound 53 in line 607, m/z 2307 [M+H-H2O-H-CH3]-loss m/z16 not m/z 349.5022 [M+H-H2O-2H-2CH3]-, referring to the loss of -H-CH3
Response: Many thanks, you improved our manuscript a lot.
- The m/z 2307 [M+H-H2O-H-CH3]-corrected to 363.2307 [M+H-2CH3-4H]+, and the m/z 349.5022 [M+H-H2O-2H-2CH3]-, corrected to 349.5022 [M+H-3CH3-3H]+
- x) lines 632-636 not in bold.
Response: The comment is made.
- y) lines 644-647 standardize compound names by starting with either an uppercase or lowercase letter, but do not mix the two.
Response: The comment is made, and components are written in lowercase.
Commented 7: Materials and methods
- a) Lines 884- 886 was exhaustively extracted with ether (40:60)- clarify 40:60 of which solventes? under laboratory conditions by soaking. What would be the conditions by soaking and for what time? The extract was concentered using a Rotary Evaporator - conditions by soaking?
Response: Purslane leaves powder (2000g) was exhaustively extracted with petroleum ether (40:60).
Several times (3 times each, 5 liters every week for three weeks).
In a lab setting at 25 oC by soaking.
A rotary evaporator was used to concentrate the extract (under reduced pressure within temperature at 35 oC.). It was then lyophilized and turned into a powder (125 g) and kept at 20 oC until use.
The statement lines 885-888.
- b) Line 889 - A549 lung cancer cell (ATCC???)
Response: ATCC: American Type Culture Collection. A549 lung cancer cells were purchased from ATCC.
- c) Lines 954-957 -Formate the sentence with the appropriate font size.
Response: The comment is made.
- d) Detail the analysis conditions of HPLC/HR-QTOF-MS/MS.
Response: We thank the reviewer for pointing this out. The comment was done, lines 1022-1026.
- e) Line 1011 Gas 1 and gas 2????
Response: Ion source gases
- References -38% of the listed references are outdated; please update them to not older than 2014.
Response: The comment is made as we explained above.
Thank you for your attention and help!
Yours Sincerely,
Samah Ali El-Newary
Associated professor at Aromatic and Medicinal Plants Department, National Research Centre,
El-Tahrir St., Dokki, Giza, 12311, Egypt.
Mobile: 01000464073

Reviewer 2 Report
Comments and Suggestions for Authors
In the manuscript (ID: ijms-3271464) studied, the phytochemical profile of Purslane petroleum ether extract and its anti-4- (methylnitrosamine)-1-(3-pyridyl) -1-but-4 none (NNK) induced lung cancer in rats were studied. Generally, the content of this manuscript meets the requirements of International Journal of Molecular Sciences. Therefore, I think that the manuscript can be accepted published in International Journal of Molecular Sciences after a major revision.
(1) Title
--Line 2-5: The title is too long and the meaning is not clear enough. Should be “The phytochemical profile of Purslane petroleum ether extract and its anti- 4-(Methylnitrosamino)-1-(3-pyridyl)-1-buta-4 none (NNK) indiced lung cancer in rats” rather “The phytochemical profile and unveiling of the promising potential of Purslane petroleum ether extract in combating lung cancer induced by 4-(Methylnitrosamino)-1-(3-pyridyl)-1-buta-4 none (NNK) in rats.”.
(2) Abstract
--Line 25: Should be “petroleum ether extract of purslane leaf” rather “purslane leaf petroleum ether extract”.
--Line 26: Should be “4-(Methylnitrosamino)-1-(3-pyridyl)-1-buta-4 none (NNK)” rather “NNK”.
--Line 26: Should be “in vitro” rather “in-vitro”. In addition, “in vitro” should be in italics. In addition, there are similar errors in other parts of the manuscript, and authors are advised to check the whole manuscript carefully and correct these minor errors.
--Line 29 and 34: Please provide the full name of ICAM and FOXO1. When an abbreviation appears in the manuscript, write its full name first, and the abbreviation is written after the full name in parentheses. Subsequently, use the abbreviation consistently and do not write out the full term again.
--Line 39: Should be “LC-MS, including” rather “LC-Mass, such as”.
(3) Keywords
--Suggest the author to add this keywords of “Purslane leaf”, “Phytochemical profile” and “lung cancer”. Besides, these keywords inducing Intercellular Adhesion Molecule-1(ICAM-1) and proliferation don't seem necessary
(4) Introduction
--Line 44: Should be “1. Introduction” rather “Introduction:”. Please read the submission requirements of International Journal of Molecular Sciences and revise the title carefully.
--In this manuscript, the review of phytochemicals to treat lung cancer lacks depth. Presently, there's a lot of research going on using phytochemicals to treat lung cancer, such as phytochemical composition and anticancer effect of Akebia trifoliata seed in non-small cell lung cancer A549 cells, Phytochemical composition and anticancer effect of Akebia trifoliata seed in non-small cell lung cancer A549 cells, etc. It is suggested that the authors systematically review the activity and structure of these ACE inhibitory peptides, so as to further explain the innovation, importance and significance of this study.
--Line 80-81: Should be “fibrosis [18].” rather “fibrosis. [18]”.
--Line 84: Should be “cancer [26].” rather “cancer.[26]”. In addition, there are similar errors in other parts of the manuscript, and authors are advised to check the whole manuscript carefully and correct these minor errors.
--Line 97: Should be “(i) unveil” rather “(i) Unveil”.
(5) Results and discussion
--Line 102-115: When we planned this study … we studied its effect on the safety profile in diseased and healthy animals.LC-QTOF-HR-MS/MS studied the metabolites of the Purslane Pet. ether extract. In the results and discussion section, the author does not need to repeat the research purpose and research method. The authors are advised to delete this section.
--Line 119, 121 and 122: Why the deepening of "A549"?
--Figure 2: Data analysis, especially significance analysis, should be applied to all experimental results of the manuscript.
--Line 116 and 132: “1.1. The Study on the A549 lung cancer cell line (in vitro investigation):” and “1.1. The study on the experimental animals (in vivo investigation):”. There is a problem with the serial number of the title. In addition, there is no serial number in the previous title, and here the serial number appears. The impression from this manuscript is that it was edited in hurry, resulting in many formatting errors and confusion.
--Figure 3B: It is suggested that all figures in this manuscript should have scales on their vertical axes, just like Figure 4B. In addition, please check the ordinate values of this figure carefully.
--Figure 11 and Table 1: Authors are advised to present key data in the main text, while other data can be used as supplementary material.
(6) Materials and methods
--1.1. Chemicals: Each material and reagents used should include Brand, catalog number, company, and the city and the country of company located.
--Line 867-877: “1.1. Chemicals” and “1.1. Authentication of the plant and extraction”. The serial number of the title in this manuscript is all wrong. It is recommended that the author carefully modify it, otherwise the manuscript cannot be accepted for publication.
--Line 888: Should be “1.1. The study on A549 lung cancer cell line (in vitro investigation)” rather “1.1. The Study on The A549 lung cancer cell line (in vitro investigation):”. It is recommended that authors carefully read the submission requirements of International Journal of Molecular Sciences, confirm whether the first letter of each word in the title needs to be capitalized, and unify the writing of the words in the title.
--Line 923: Should be “(20–25 °C” rather “(20–25 Cο”.
Comments on the Quality of English LanguageIt is recommended that the author carefully modify it, otherwise the manuscript cannot be accepted for publication.
Author Response
Author’s Response
Responses to reviewers’ comments for “The phytochemical profile and unveiling of the promising potential of Purslane petroleum ether extract in combating lung cancer induced by 4-(Methylnitrosamino)-1-(3-pyridyl)-1-butanone (NNK) in rats.” (Manuscript ID: ijms-3271464).
Dear Editor,
We thank the reviewers for their insightful comments, which enabled us to improve the manuscript. We hope that you will accept it after revision. According to the reviewers' suggestions, the manuscript has been revised carefully. The detailed corrections raised by the reviewers have been addressed point by point and highlighted in yellow in the revised manuscript.
Reviewer 2
In the manuscript (ID: ijms-3271464) studied, the phytochemical profile of Purslane petroleum ether extract and its anti-4- (methylnitrosamine)-1-(3-pyridyl) -1-but-4 none (NNK) induced lung cancer in rats were studied. Generally, the content of this manuscript meets the requirements of International Journal of Molecular Sciences. Therefore, I think that the manuscript can be accepted published in International Journal of Molecular Sciences after a major revision.
Response: A lot of thanks to the reviewer for this comment.
Commented 1: (1) Title
Line 2-5: The title is too long and the meaning is not clear enough. Should be “The phytochemical profile of Purslane petroleum ether extract and its anti- 4-(Methylnitrosamino)-1-(3-pyridyl)-1-buta-4 none (NNK) indiced lung cancer in rats” rather “The phytochemical profile and unveiling of the promising potential of Purslane petroleum ether extract in combating lung cancer induced by 4-(Methylnitrosamino)-1-(3-pyridyl)-1-buta-4 none (NNK) in rats.”.
Response 1: We thank the reviewer for his comments, which enabled us to improve our work. We changed the title to “The phytochemical profile of Purslane leaves petroleum ether extract and its anticancer effect on 4-(Methylnitrosamino)-1-(3-pyridyl)-1-buta-4 none (NNK) induced lung cancer in rats.”
Commented 2: (2) Abstract
- Line 25: Should be “petroleum ether extract of purslane leaf” rather “purslane leaf petroleum ether extract”.
Response: Comment is made, line 24.
- Line 26: Should be “4-(Methylnitrosamino)-1-(3-pyridyl)-1-buta-4 none (NNK)” rather “NNK”.
Response: Comment is made.
- Line 26: Should be “in vitro” rather “in-vitro”. In addition, “in vitro” should be in italics. In addition, there are similar errors in other parts of the manuscript, and authors are advised to check the whole manuscript carefully and correct these minor errors.
Response: Comment is made throughout the manuscript.
- Line 29 and 34: Please provide the full name of ICAM and FOXO1. When an abbreviation appears in the manuscript, write its full name first, and the abbreviation is written after the full name in parentheses. Subsequently, use the abbreviation consistently and do not write out the full term again.
Response: Comment is made, and all abbreviations in the abstract are written in full names.
- Line 39: Should be “LC-MS, including” rather “LC-Mass, such as”.
Response: Comment is made.
Commented 3: (3) Keywords
Suggest the author to add this keywords of “Purslane leaf”, “Phytochemical profile” and “lung cancer”. Besides, these keywords inducing Intercellular Adhesion Molecule-1(ICAM-1) and proliferation don't seem necessary
Response 3: Comment is made.
Commented 4: (4) Introduction
- Line 44: Should be “1. Introduction” rather “Introduction:”. Please read the submission requirements of International Journal of Molecular Sciences and revise the title carefully.
Response: Comment is made through the manuscript.
- In this manuscript, the review of phytochemicals to treat lung cancer lacks depth. Presently, there's a lot of research going on using phytochemicals to treat lung cancer, such as phytochemical composition and anticancer effect of Akebia trifoliata seed in non-small cell lung cancer A549 cells, Phytochemical composition and anticancer effect of Akebia trifoliata seed in non-small cell lung cancer A549 cells, etc. It is suggested that the authors systematically review the activity and structure of these ACE inhibitory peptides, so as to further explain the innovation, importance and significance of this study.
Response
- Line 80-81: Should be “fibrosis [18].” rather “fibrosis. [18]”.
Response: Comment is made.
- Line 84: Should be “cancer [26].” rather “cancer. [26]”. In addition, there are similar errors in other parts of the manuscript, and authors are advised to check the whole manuscript carefully and correct these minor errors.
Response: Comment is made.
- Line 97: Should be “(i) unveil” rather “(i) Unveil”.
Response: Comment is made.
Commented 5: (5) Results and discussion
- Line 102-115: When we planned this study … we studied its effect on the safety profile in diseased and healthy animals.LC-QTOF-HR-MS/MS studied the metabolites of the Purslane Pet. ether extract. In the results and discussion section, the author does not need to repeat the research purpose and research method. The authors are advised to delete this section.
Response: The paragraph is removed from the manuscript.
- Line 119, 121 and 122: Why the deepening of "A549"?
Response: The deepening is removed.
- Figure 2: Data analysis, especially significance analysis, should be applied to all experimental results of the manuscript.
Response: A significance analysis was added in Figure 2 of the manuscript.
- Line 116 and 132: “1.1. The Study on the A549 lung cancer cell line (in vitro investigation):” and “1.1. The study on the experimental animals (in vivo investigation):”. There is a problem with the serial number of the title. In addition, there is no serial number in the previous title, and here the serial number appears. The impression from this manuscript is that it was edited in hurry, resulting in many formatting errors and confusion.
Response: The serial numbers of titles are corrected throughout the manuscript.
- Figure 3B: It is suggested that all figures in this manuscript should have scales on their vertical axes, just like Figure 4B. In addition, please check the ordinate values of this figure carefully.
Response: The comment is made.
- Figure 11 and Table 1: Authors are advised to present key data in the main text, while other data can be used as supplementary material.
Response: Dear reviewer
Please keep this section because we work in a team work and each of us has his own specialty. In this specialty, three authors work in the field of plant phytochemistry, and this research is presented to promote each in his field; they played this role in the manuscript because their specialty is to interpret, define, and identify the polyphenolic components using HPLC-QTOF-MS/MS. This section includes evidence and explanations for the presence of these anti-lung cancer compounds. If this section is deleted, this means that this research team did not cooperate in this work and thus cancels the role that they originally performed. Please understand the problem that will return to the team of this specialty that performed the work seriously for this section and please, accept the existence of this section in the manuscript as we are grateful to you.
Commented 6: (6) Materials and methods
- 1. Chemicals: Each material and reagents used should include Brand, catalog number, company, and the city and the country of company located.
Response: Brand, company, and the city and the country of company located of all kits and chemical are places. Lines 867-877 and 972-979.
- Line 867-877: “1.1. Chemicals” and “1.1. Authentication of the plant and extraction”. The serial number of the title in this manuscript is all wrong. It is recommended that the author carefully modify it, otherwise the manuscript cannot be accepted for publication.
Response: The serial numbers of titles are corrected throughout the manuscript.
- Line 888: Should be “1.1. The study on A549 lung cancer cell line (in vitro investigation)” rather “1.1. The Study on The A549 lung cancer cell line (in vitro investigation):”. It is recommended that authors carefully read the submission requirements of International Journal of Molecular Sciences, confirm whether the first letter of each word in the title needs to be capitalized, and unify the writing of the words in the title.
Response: Comment is made throughout the manuscript.
- Line 923: Should be “(20–25 °C” rather “(20–25 Cο”.
Response: Comment is made.
Thank you for your attention and help!
Yours Sincerely,
Samah Ali El-Newary
Associated professor at Aromatic and Medicinal Plants Department, National Research Centre,
El-Tahrir St., Dokki, Giza, 12311, Egypt.
Mobile: 01000464073

Reviewer 3 Report
Comments and Suggestions for Authors
The manuscript entitled “The phytochemical profile and unveiling of the promising po- 2 tential of Purslane petroleum ether extract in combating lung 3 cancer induced by 4-(Methylnitrosamino)-1-(3-pyridyl)-1-buta- 4 none (NNK) in rats” describes the ability of purslane leaves petroleum ether extract to treat lung cancer.
Minor changes are proposed. Below you can find my comments:
o Line 63 and 954: the sentence is not with the proper size (check throughout the manuscript)
o Line 101: Transfer “Materials and Methods” before “Results and discussion”
o Line 125-129: increase the font size
o Line 476: analouges or analogues
o Line 597: subscript in the number 2 in H2O
o Line 702: “Indeed, In A549…” the I as subscript
o Line 711: “….non-non-small-cell…” non is double?
o Line 711: The in vitro or in vivo should either italics or not throughout the manuscript
o Line 883 and 887: the degrees are missing from the temperature
o Line 894: space between invitro
o Line 1002: Replace CAN with ACN
The supplementary material was submitted?
Author Response
Author’s Response
Responses to reviewers’ comments for “The phytochemical profile and unveiling of the promising potential of Purslane petroleum ether extract in combating lung cancer induced by 4-(Methylnitrosamino)-1-(3-pyridyl)-1-butanone (NNK) in rats.” (Manuscript ID: ijms-3271464).
Dear Editor,
We thank the reviewers for their insightful comments, which enabled us to improve the manuscript. We hope that you will accept it after revision. According to the reviewers' suggestions, the manuscript has been revised carefully. The detailed corrections raised by the reviewers have been addressed point by point and with blue color and highlighted in yellow in the revised manuscript.
Reviewer 3:
Minor changes are proposed. Below you can find my comments:
- Line 63 and 954: the sentence is not with the proper size (check throughout the manuscript)
- Response: The comment is made.
- Line 101: Transfer “Materials and Methods” before “Results and discussion”
- Response: This is the journal format.
- Line 125-129: increase the font size
- Response: The font is corrected throughout the manuscript.
- Line 476: analouges or analogues
- Response: Analogues corrected to analogues.
- Line 597: subscript in the number 2 in H2O
- Response: The comment is made.
- Line 702: “Indeed, In A549…” the I as subscript
- Response: The comment is made.
- Line 711: “….non-non-small-cell…” non is double?
- Response: The comment is made.
- Line 711: The in vitro or in vivo should either italics or not throughout the manuscript
- Response: The comment is made throughout the manuscript.
- Line 883 and 887: the degrees are missing from the temperature
- Response: The comment is made.
- Line 894: space between invitro
- Response: The comment is made.
- Line 1002: Replace CAN with CAN.
- Response: The comment was made.
Thank you for your attention and help!
Yours Sincerely,
Samah Ali El-Newary
Associated professor at Aromatic and Medicinal Plants Department, National Research Centre,
El-Tahrir St., Dokki, Giza, 12311, Egypt.
Mobile: 01000464073

Reviewer 4 Report
Comments and Suggestions for Authors
Authors carried out phytochemical profile and Purslane petroleum ether extract in combating lung cancer model
Minor issues
1. Manuscripts analytics data is too dense and needs to be condensed to the essentials. Table1. should be put in supplement.
2. Unify the font and size of all pictures (Write according to the publisher's rules).
3. The arrangement in Figure 1, 10 and 11 needs to be adjusted.
4. Figure 7. 8 and 9 use arrows to be specific to make it easier for the reader to understand.
5. Follow the publisher's rules for bold and italics in sentences, etc.
6. Remove the blue color from the sentence
Major issues
1. More research is needed to determine which of the various substances isolated have efficacy in lung cancer.
2. Authors should include RT-PCR or Western results for mRNA and protein analysis for biomarkers.
3. The author must reorganize and rewrite the manuscript in its current format.
Author Response
Author’s Response
Responses to reviewers’ comments for “The phytochemical profile and unveiling of the promising potential of Purslane petroleum ether extract in combating lung cancer induced by 4-(Methylnitrosamino)-1-(3-pyridyl)-1-butanone (NNK) in rats.” (Manuscript ID: ijms-3271464).
Dear Editor,
We thank the reviewers for their insightful comments, which enabled us to improve the manuscript. We hope that you will accept it after revision. According to the reviewers' suggestions, the manuscript has been revised carefully. The detailed corrections raised by the reviewers have been addressed point by point and highlighted in yellow in the revised manuscript.
Reviewer 4
Authors carried out phytochemical profile and Purslane petroleum ether extract in combating lung cancer model.
Minor issues
- Manuscripts analytics data is too dense and needs to be condensed to the essentials. Table1. should be put in supplement.
Response: Dear reviewer
Please keep this section because we work in a team work and each of us has his own specialty. In this specialty three authors work in the field of plant phytochemistry and this research is presented to promote each in his field; and they played this role in the manuscript because their specialty is to interpret, define and identify the polyphenolic components using HPLC-QTOF-MS/MS. This section includes evidences and explanations for the presence of these anti-lung cancer compounds. If this section is deleted, this means that this research team did not cooperate in this work and thus cancels their role that they originally performed. Please understand the problem that will return to the team of this specialty that performed the work seriously for this section and please, accept the existence of this section in the manuscript as we are grateful to you.
- Unify the font and size of all pictures (Write according to the publisher's rules).
Response: The comment is made.
- The arrangement in Figure 1, 10 and 11 needs to be adjusted.
Response: The comment was made in blue color.
- Figure 7. 8 and 9 use arrows to be specific to make it easier for the reader to
understand.
Response: The comment is made.
- Follow the publisher's rules for bold and italics in sentences, etc.
Response: The comment is made. We removed bold words and made italic words like in vitro and in vivo.
- Remove the blue color from the sentence
Response: The comment is made.
Major issues
Commented 1: More research is needed to determine which of the various substances isolated have efficacy in lung cancer.
Response 1: It was determined that isolated substances were active by measuring in vitro cytotoxicity activity on A549 lung cancer cells and determining more research for these isolated substances in further study.
Commented 2: Authors should include RT-PCR or Western results for mRNA and protein analysis for biomarkers.
Response 2: The manuscript already included RT-PCR results for FOXO1, MMP9, and IGF-1 genes in Quantitative real-time PCR section in methods.
Commented 3: The author must reorganize and rewrite the manuscript in its current format.
Response 2: The comment is made.
Thank you for your attention and help!
Yours Sincerely,
Samah Ali El-Newary
Associated professor at Aromatic and Medicinal Plants Department, National Research Centre,
El-Tahrir St., Dokki, Giza, 12311, Egypt.
Mobile: 01000464073

Round 2
Reviewer 1 Report
Comments and Suggestions for Authors
The article shows the phytochemical characterization of Purslane petroleum ether extract and its potential in combating lung cancer induced by 4-(methylnitrosamino)-1-(3-pyridyl)-1-butane-4-nonone (NNK) in rats. The article has been improved and can be accepted in its current form.
Author Response

(The authors gave the same response as above.)

Reviewer 2 Report
Comments and Suggestions for Authors
The manuscript (ijms-3271464) researched the phytochemical profile and promising potential on lung cancer induced by 4-(Methylnitrosamino)-1-(3-pyridyl)-1-butanone (NNK) in rats of Purslane petroleum ether extract。 Generally, the contents meet the requirements of International Journal of Molecular Sciences. Moreover, the authors have answered my questions well and made the necessary changes to the manuscript. Then, I think that the manuscript can be accepted for publication in International Journal of Molecular Sciences.
Author Response

(The authors gave the same response as above.)

Reviewer 3 Report
Comments and Suggestions for Authors
The revised version of the manuscript is ready for publication.
Author Response

(The authors gave the same response as above.)

Reviewer 4 Report
Comments and Suggestions for Authors
Well done
Author Response

(The authors gave the same response as above.)
